# Adaptive Window Pruning for Efficient Local Motion Deblurring

**Haoying Li**[1,2†]**, Jixin Zhao**[1]**, Shangchen Zhou**[1]**,**
**Huajun Feng**[2]**, Chongyi Li**[3†*] **& Chen Change Loy**[1]
[1]S-Lab, Nanyang Technological University, Singapore
[2]Zhejiang University, China
[3]Nankai University, China
{lhaoying, fenghj}@zju.edu.cn
{ZHAO0388, sczhou, ccloy}@ntu.edu.sg
{lichongyi}@nankai.edu.cn

## Abstract

Local motion blur commonly occurs in real-world photography due to the mixing between moving objects and stationary backgrounds during exposure. Existing image deblurring methods predominantly focus on global deblurring, inadvertently affecting the sharpness of backgrounds in locally blurred images and wasting unnecessary computation on sharp pixels, especially for high-resolution images. This paper aims to adaptively and efficiently restore high-resolution locally blurred images. We propose a local motion deblurring vision Transformer (LMD-ViT) built on adaptive window pruning Transformer blocks (AdaWPT). To focus deblurring on local regions and reduce computation, AdaWPT prunes unnecessary windows, only allowing the active windows to be involved in the deblurring processes. The pruning operation relies on the blurriness confidence predicted by a confidence predictor that is trained end-to-end using a reconstruction loss with Gumbel-Softmax re-parameterization and a pruning loss guided by annotated blur masks. Our method removes local motion blur effectively without distorting sharp regions, demonstrated by its exceptional perceptual and quantitative improvements compared to state-of-the-art methods. In addition, our approach substantially reduces FLOPs by 66% and achieves more than a twofold increase in inference speed compared to Transformer-based deblurring methods. Codes and data are available at https://leiali.github.io/LMD-ViT_webpage.

## 1 Introduction

Contrary to global motion blur, which typically affects an entire image (Zhang et al., 2018), local motion blur is confined to specific regions within an image. Such local blur is generally the result of object movements captured by stationary cameras (Li et al., 2023; Schelten & Roth, 2014). Applying global deblurring methods to images featuring local motion blur inevitably introduces unwanted distortions in regions that were originally sharp, as illustrated in Figure 1. Moreover, the processing of sharp regions, which is not required in this context, leads to unnecessary computational expenditure. This wastage becomes particularly noticeable when dealing with high-resolution inputs. Existing local motion deblurring methods, such as LBAG (Li et al., 2023), address the issue by detecting local blur regions for targeted processing. While LBAG uses a gate structure to mitigate the deblurring impact on non-blurred regions, it still involves unnecessary computations as the entire image is processed by the network. In addition, the method's reliance on a Convolutional Neural Network (CNN) architecture leads to limitations, as the interactions between the image and convolution kernels are content-independent and ill-suited for modeling long-range dependencies. The Transformer architecture (Vaswani et al., 2017), which excels at long-range pixel interactions, has been successfully applied in several image restoration problems (Liang et al., 2021a; Guo et al.,

---

[†]This paper was completed in S-Lab, Nanyang Technological University.
[*]Corresponding author.

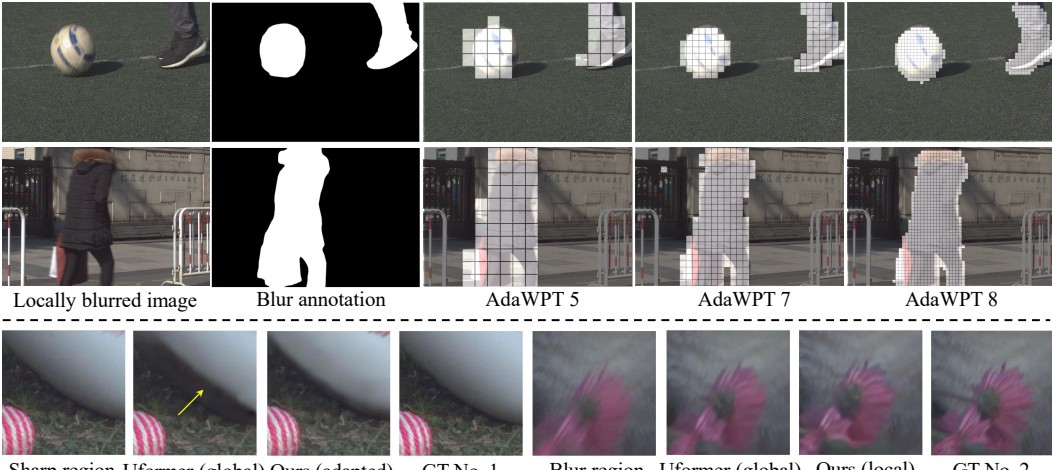

| Locally blurred image | Blur annotation | AdaWPT 5 | AdaWPT 7 | AdaWPT 8 |

| Sharp region | Uformer (global) | Ours (adapted) | GT No. 1 | Blur region | Uformer (global) | Ours (local) | GT No. 2 |

Figure 1: We introduce LMD-ViT, a Transformer-based local motion deblurring method with an adaptive window pruning mechanism. We prune unnecessary windows based on the predicted blurriness confidence supervised by our blur region annotation. In this process, the feature maps are pruned at varying levels of granularity within blocks of different resolutions. The white masks in AdaWPT 5 to 8 denote tokens to be preserved, and regions without white masks are pruned. The grids denote window borders. Unlike global deblurring methods that modify global regions (Wang et al., 2022; Zamir et al., 2022), LMD-ViT performs dense computing only on the active windows of blurry regions. Consequently, local blurs are efficiently removed without distorting sharp regions.

2023; Wang et al., 2022; Zamir et al., 2022). However, Transformers for deblurring tasks (Wang et al., 2022; Zamir et al., 2022) usually conduct attention operations on every token, which not only distorts sharp tokens in locally blurry images (as shown in Figure 1) but also require substantial memory and extend inference time. To mitigate these computational demands, strategies such as channel inter-dependence operations (Liang et al., 2021b), window-division (Wang et al., 2022; Yang et al., 2021) and token-reducing (Bolya et al., 2022; Liang et al., 2022; Meng et al., 2022; Rao et al., 2021; Yin et al., 2022) have been proposed.

Drawing inspiration from preceding work, in this paper, we propose a U-shaped local motion deblurring vision Transformer (LMD-ViT) with adaptive window pruning Transformer blocks (AdaWPT) as its core component. Our proposed LMD-ViT is the first to apply sparse vision Transformer to the local motion deblurring task. The core block, AdaWPT, focuses on locally blurred regions rather than global regions, which is made possible by removing windows unrelated to blurred areas, surpassing prior state-of-the-art methods in both deblurring performance and efficiency. Specifically, we first train a confidence predictor which is able to automatically predict theblurriness confidenceof feature maps. It is trained end-to-end by a reconstruction loss with Gumbel-Softmax reparameterization, and a pruning loss guided by our elaborately annotated local blur masks. We then design a decision layer that provides binary decision maps in which "1" represents the kept tokens in blur-related regions that require processing during inference while "0" represents the abandoned tokens in the other regions that can be removed. We also propose a window pruning strategy with Transformer layers. In detail, we apply window-based multi-head self-attention (W-MSA) (Wang et al., 2022) and window-based feed-forward layers rather than enforcing these Transformer layers globally. Only the selected windows are forwarded to these window-based Transformer layers, preventing unnecessary distortion of sharp regions while also reducing computations. To further enhance content interactions, AdaWTP employs shifted window mechanism (Liang et al., 2021a; Liu et al., 2021) and position embeddings (Wang et al., 2022) among the Transformer layers. Furthermore, we insert AdaWTP in LMD-ViT under different scales and receptive fields. Therefore, AdaWTP conducts coarse pruning of windows in low-resolution layers and more refined pruning in high-resolution layers, as shown in Figure 1, which achieves a balance between computational complexity and deblurring performance. Compared to the CNN-based local deblurring methods, our proposed method achieves improved visual and quantitative performances. Compared to Transformer-based local deblurring methods, we speed up inference.

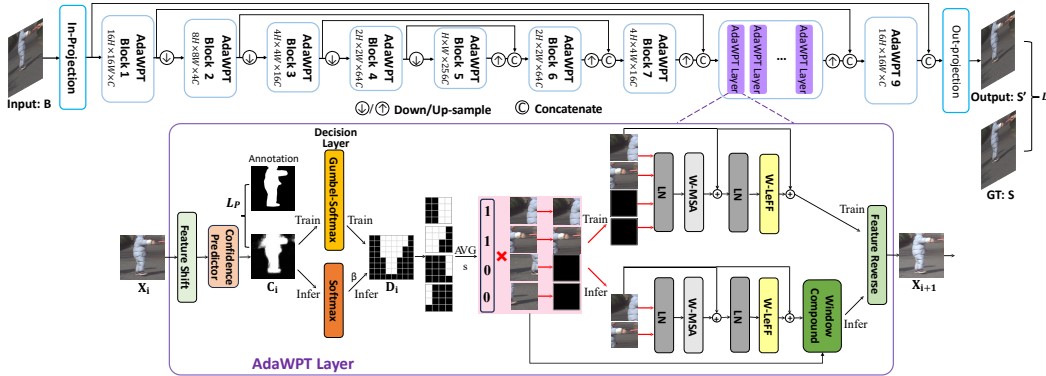

Figure 2: The architecture of LMD-ViT. LMD-ViT is built on a U-shape encoder-decoder structure with AdaWPT blocks, which prune tokens at different resolutions. Each AdaWPT block contains several AdaWPT layers, predicting the blurriness confidence ($C_i$), the pruning decision ($D_i$) and calculating the pruning loss ($\mathcal{L}_P$). $X_i$ and $X_{i+1}$ denote the $i^{th}$ and $i + 1^{th}$ feature map of each layer, respectively. "AVG", "$\beta$", "s" and "$\mathcal{L}_R$" denote the average pooling operation, the pruning threshold in inference, the threshold of AVG, and the reconstruction loss respectively.

To summarize, our main contributions are: 1) the first sparse vision Transformer framework for local motion deblurring, LMD-ViT, which utilizes an adaptive window pruning strategy to focus computation on localized regions affected by blur and achieve efficient blur reduction without causing unnecessary distortion to sharp regions; 2) a sophisticated blurriness prediction mechanism, integrating a confidence predictor and decision layer to effectively distinguish between sharp and blurry regions; 3) carefully annotated local blur masks for ReLoBlur dataset (Li et al., 2023), which improve the performance of local deblurring methods. We explain more background information of our method in Appendix A.

## 2 METHODOLOGY

### 2.1 MODEL ARCHITECTURE

The architecture of our local motion deblurring vision Transformer (LMD-ViT) is shown at the top of Figure 2, which is a U-shaped network with an encoder stage, a bottleneck stage, and a decoder stage with skip connections. An in-projection/out-projection layer is placed at the beginning/end of the network to extract RGB images to feature maps or convert feature maps to RGB images. The encoder, bottleneck, and decoder include a series of adaptive window-token pruning Transformer blocks (AdaWPT) and down-sampling/up-sampling layers. As a key component, AdaWPT removes local blurs by a window pruning strategy with a confidence predictor, a decision layer, and several Transformer layers. It is trained with a reconstruction loss ($\mathcal{L}_R$) and a pruning loss ($\mathcal{L}_P$) constrained by our carefully annotated blur masks. AdaWPT can be applied in any encoder/decoder/bottom-neck block and is flexible to prune at different resolutions. As shown in Figure 1, in our proposed LMD-ViT, windows are pruned coarsely in low-resolution blocks and finely in high-resolution blocks. This strikes a balance between computational complexity and accuracy. We detail the architectures and model hyper-parameters of LMD-ViT in Appendix B.

### 2.2 ADAPTIVE WINDOW PRUNING TRANSFORMER (ADAWPT)

As shown at the bottom of Figure 2, an adaptive window pruning Transformer block (AdaWPT) includes several AdaWPT layers. During training, each AdaWPT layer comprises a confidence predictor, a decision layer, a feature shift/reverse block, and several Transformer layers such as window-based multi-head self-attention (W-MSA) (Wang et al., 2022), window-based locally-enhanced feed-forward layer (W-LeFF), and layer normalization (LN). AdaWPT is responsible for predicting the blurriness confidence by a confidence predictor, generating pruning decisions from a decision layer and pruning windows. It only allows the unpruned windows to be fed into the window-based Transformer layers, significantly reducing computational costs and keeping sharp regions undistorted.

Besides, a feature shift/reverse block is inserted before/after pruning to promote feature interactions. In the inference phase, a window compound layer is incorporated to integrate the abandoned windows and selected windows into a unified feature map. We detail the modules in AdaWPT in the following paragraphs.

### 2.3 CONFIDENCE PREDICTOR

The confidence predictor predicts the blurriness confidence for the input features $\mathbf{X_i} \in \mathbb{R}^n$. Tokens with higher confidence are more likely to be kept and others are removed. Following Rao et al. (2021), the confidence predictor employs MLP layers to produce feature embeddings $e$ and predict the confidence map $\mathbf{C}$ using Softmax:

$$\mathbf{C_i} = \text{Softmax}(e_i),\ e_i = \text{Concat}\left(\text{MLP}\left(\text{MLP}(\mathbf{X_i}), \frac{\sum_{j=n}^{N} \mathbf{D}_j \cdot \text{MLP}(\mathbf{X_{i_j}})}{\sum_{j=n}^{N} \mathbf{D}_j}\right)\right), i \in \mathbb{N}_+, \tag{1}$$

where $\mathbf{D}$, initialized using $\mathbf{I}$, is the one-hot decision map to prune windows, which will be introduced in Section 2.4. $i$ denotes the $i^{th}$ AdaWPT block. The confidence predictor is trained using an end-to-end reconstruction loss (introduced in Section 2.7) with Gumbel-Softmax parameterization (see Section 2.4), together with a pruning loss (introduced in Section 2.7) guided by our blurry mask annotations (introduced in Section 2.6).

### 2.4 DECISION LAYER

The decision layer samples from the blurriness confidence map to generate binary pruning decisions $\mathbf{D}$, in which "1" represents tokens to be kept while "0" represents tokens to be abandoned. Although our goal is to perform window pruning, it is not practical to remove the zero tokens directly, for the absence of tokens halts backward propagation, and the different removal instances make parallel computing impossible in end-to-end training. To overcome this issue, we design the decision layer for training and testing, respectively.

In training, we apply the Gumbel-Softmax re-parameterization (Jang et al., 2017) as the decision layer, since it assures the gradients to flow through the network when sampling the training decision map $\mathbf{D^{tr}}$ from the training confidence map $\mathbf{C^{tr}}$:

$$\mathbf{D_i^{tr}} = \text{Gumbel-Softmax}(\mathbf{C_i^{tr}}). \tag{2}$$

In testing, we apply Softmax with a constant threshold $\beta$ as the decision layer:

$$\mathbf{D_i^{te}}(x,y) = \begin{cases} 0, & \text{Softmax}\big(\mathbf{C^{te}}_i(x,y))\big) < \beta \\ 1, & \text{Softmax}\big(\mathbf{C^{te}}_i(x,y))\big) \geq \beta, \end{cases} \tag{3}$$

where $(x,y)$ denotes a coordinate point. The abandoned windows, that is, windows irrelevant to local blurs, are further set to zero by

$$\mathbf{X_i'} = \mathbf{D_i} \cdot \mathbf{X_i}. \tag{4}$$

### 2.5 EFFICIENT TRANSFORMER LAYERS WITH WINDOW PRUNING STRATEGY

Considering the quadratic computation costs with respect to a large number of tokens of high-resolution images, we employ Transformer layers in non-overlapping windows rather than in a global manner to accelerate training and inference. Specifically, we choose W-MSA layer (Wang et al., 2022) for the self-attention (SA) operation. For the feed-forward structure, we develop a window-based locally-enhanced feed-forward layer (W-LeFF) as an alternative to LeFF (Wang et al., 2022) which modifies global tokens. W-LeFF uses a $3 \times 3$ convolution with stride 1 and reflected padding 1 in independent windows. The reflected padding ensures the continuity at the window borders and obtains almost the same performance as LeFF (Wang et al., 2022) (we compare the performance of LeFF (Wang et al., 2022) and W-LeFF in Appendix E.1).

To enable parallel computing, we regard each window as a token group and prune based on windows. Each window owns a 0/1 decision, which is calculated by average pooling the decision map with a threshold $s$. Windows with an average $\geq s$ are regarded as the kept windows and the others as the

abandoned windows. To promote content interactions among non-overlapping windows, we also apply relative position encoding (Wang et al., 2022) in the attention module and shift/reverse the windows by half the window size at the beginning/end of the AdaWTP.

To accomplish both differentiable training and fast inference, we propose different pruning strategies with W-MSA and W-LeFF in training and testing, respectively. In training, to ensure back-propagation, all the windows including the abandoned windows go through W-MSA, W-LeFF, and LN sequentially to generate training features $\mathbf{X_{i+1}^t}$:

$$\mathbf{X_{i+1}^{tr}}' = \text{W-MSA}(\text{LN}(\mathbf{X_i^{tr}}')) + \mathbf{X_i^{tr}}', \ \mathbf{X_{i+1}^{tr}} = \text{W-LeFF}(\text{LN}(\mathbf{X_{i+1}^{tr}}')) + \mathbf{X_{i+1}^{tr}}'. \tag{5}$$

In testing, only the kept windows are processed by Transformer layers, which release a great number of unnecessary tokens to deblur. To enable future stages to perform global operations, we mend the abandoned windows to their original locations to compound a complete testing feature map $\mathbf{X_{i+1}^{te}}$ :

$$\mathbf{X_{i+1}^{te}}' = \text{W-MSA}(\text{LN}(\mathbf{X_i^{te}}' \geq s)) + \mathbf{X_i^{te}}', \ \mathbf{X_{i+1}^{te}} = \text{W-LeFF}(\text{LN}(\mathbf{X_{i+1}^{te}}' \geq \mathbf{s})) + \mathbf{X_{i+1}^{te}}', \tag{6}$$

where $s$ denotes the threshold in the average pooling operation (AVG).

## 2.6 Blur region annotation

To obtain the ground-truth local blur mask for supervised training of the confidence predictor, we carefully annotate local blur masks for the ReLoBlur dataset (Li et al., 2023). Firstly, we roughly identify the blurry regions by subtracting the blurry image and its corresponding sharp image to observe the difference. This difference map helps us estimate the location of blurriness. Subsequently, we utilize *EISeg* software[1] to isolate moving objects and automatically segment their components. We then undertake manual refinement, guided by a principle: any diffuse spot encompassing more than 5 pixels is categorized as blurry, while those encompassing fewer are classified as sharp. To mitigate any potential leakage, we extend the blurry region by 5 pixels to ensure comprehensive coverage within the annotated mask. In our annotated blur masks, a pixel value of 1 denotes blurriness, while a value of 0 signifies sharpness, as shown in Figure 1 and Figure 6.

## 2.7 Loss functions

To guide adaptive window pruning and locally deblurring, we propose a pruning loss and combine it with a weighted reconstruction loss to form the total loss:

$$\mathcal{L} = \mathcal{L}_{\mathcal{P}} + \mathcal{L}_{\mathcal{R}}. \tag{7}$$

**Pruning loss.** We propose a pruning loss to constrain the blurriness confidence prediction:

$$\mathcal{L}_{\mathcal{P}} = \lambda_0 \sum_{i=1}^{n} \text{Cross-Entropy}(\mathbf{C_i}, \text{Down-Sample}^i(\mathbf{M})), \tag{8}$$

where $\lambda_0$=0.01, and the blur mask $\mathbf{M}$ is down-sampled to match the resolution of the confidence maps, and $i$ indicates a confidence predictor's location.

**Reconstruction loss.** To focus training on local blurred regions, following Li et al. (2023), we apply weights $w$ on the reconstruction loss, which is a combination of $L_1$ loss, SSIM loss, and FFT loss:

$$\mathcal{L}_{\mathcal{R}} = w\mathcal{L}_{\mathcal{R}}'(\mathbf{M} \cdot \mathbf{S}', \mathbf{M} \cdot \mathbf{S}) + (1-w)\mathcal{L}_{\mathcal{R}}'((\mathbf{1} - \mathbf{M}) \cdot \mathbf{S}', (\mathbf{1} - \mathbf{M}) \cdot \mathbf{S}),$$
$$\mathcal{L}_{\mathcal{R}}' = L_1(\mathbf{S}', \mathbf{S}) + \lambda_2 SSIM(\mathbf{S}', \mathbf{S}) + \lambda_3 FFT(\mathbf{S}', \mathbf{S}), \tag{9}$$

where $S$, and $S'$ denote the sharp ground truth, and the deblurred output. $w = 0.8$, $\lambda_1$=$\lambda_2$=1.0, and $\lambda_3$=0.1 denote the weights of the loss functions.

# 3 Experiments and analyses

## 3.1 Experimental settings

We train LMD-ViT using AdamW optimizer (Kingma & Ba, 2015) with the momentum terms of (0.9, 0.999), a batch size of 12, and an initial learning rate of $2 \times 10^{-4}$ that is updated every 2k steps

---

[1]*EISeg* is provided by https://github.com/PaddlePaddle/PaddleSeg/tree/release/2.7/EISeg5

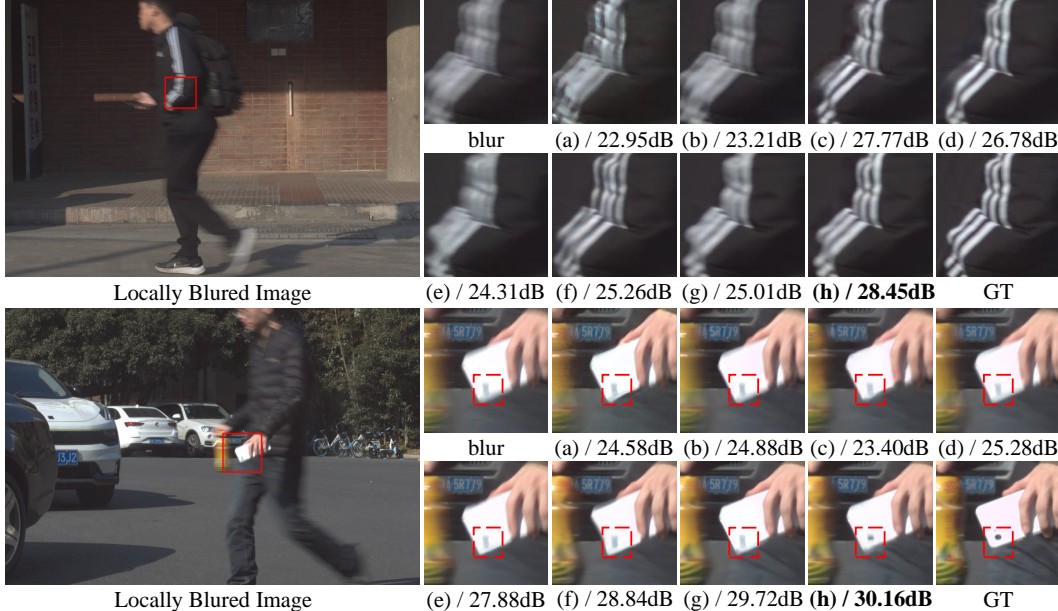

blur (a) / 22.95dB (b) / 23.21dB (c) / 27.77dB (d) / 26.78dB

Locally Blurred Image (e) / 24.31dB (f) / 25.26dB (g) / 25.01dB (**h**) / **28.45dB** GT

blur (a) / 24.58dB (b) / 24.88dB (c) / 23.40dB (d) / 25.28dB

Locally Blurred Image (e) / 27.88dB (f) / 28.84dB (g) / 29.72dB (**h**) / **30.16dB** GT

Figure 3: Visual comparisons of state-of-the-art methods for local motion deblurring. (a) DeepDeblur (Nah et al., 2017); (b) DeblurGAN_v2 (Kupyn et al., 2019); (c) HINet (Chen et al., 2021); (d) MIMO-UNet (Cho et al., 2021); (e) LBAG (Li et al., 2023); (f) Restormer (Zamir et al., 2022); (g) Uformer (Wang et al., 2022); (h) LMD-ViT (ours).

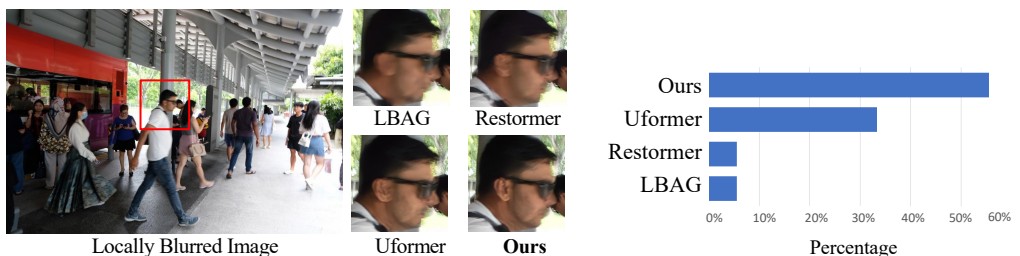

Figure 4: Results of user study for real-world local motion deblurring. The horizontal axis indicates the percentage of favoring each method.

by a cosine annealing schedule (Loshchilov & Hutter, 2017). We set the window size of AdaWPT to $8 \times 8$, and the initial embedded dim to 32 which is doubled after passing each down-sampling layer. LMD-ViT is trained on the GoPro dataset (Nah et al., 2017) and ReLoBlur dataset (Li et al., 2023) together, in order to enable both local and global motion deblurring. The sampling ratio of the GoPro training data (Nah et al., 2017) and the ReLoBlur training data (Li et al., 2023) is close to 1:1. For a fair comparison, we train the baseline methods using the same datasets and cropping strategy. The model configurations of the compared deblurring methods follow their origin settings.

We evaluate our proposed LMD-ViT and baseline methods on the ReLoBlur testing dataset (Li et al., 2023) with the full image size of 2152×1436 on 1 Nvidia A100 GPU. In testing, the inputs are solely of locally blurred images without accompanying blur masks. In addition to the commonly-used PSNR and SSIM (Wang et al., 2004) metrics, we calculate weighted PSNR and weighted SSIM (Li et al., 2023) to better assess the local deblurring performance. We provide the evaluation results in the following sections and Appendix F.

## 3.2 EXPERIMENTAL RESULTS

**Evaluations on public datasets.** We first compare the proposed LMD-ViT with both CNN-based methods (Nah et al., 2017; Kupyn et al., 2019; Chen et al., 2021; Cho et al., 2021; Li et al., 2023)

Table 1: Quantitative comparisons on the local deblurring dataset. "$PSNR_w$", "$SSIM_w$", "Time" and "Params" denote weighted PSNR, weighted SSIM, inference time, and model parameters respectively. We bold the best result under each evaluation metric.

| Categories | Methods | ↑PSNR | ↑SSIM | ↑$PSNR_w$ | ↑$SSIM_w$ | Time | Params | FLOPs |
|---|---|---|---|---|---|---|---|---|
| CNNs | DeepDeblur (Nah et al., 2017) | 33.02 | 0.8904 | 28.29 | 0.8398 | 0.50s | 11.72M | 17.133T |
| | DeblurGAN-v2 (Kupyn et al., 2019) | 33.26 | 0.8975 | 28.29 | 0.8489 | 0.07s | 5.076M | 0.989T |
| | HINet (Chen et al., 2021) | 34.40 | 0.9160 | 28.82 | 0.8672 | 0.31s | 88.67M | 8.696T |
| | MIMO-UNet (Cho et al., 2021) | 34.64 | 0.9247 | 29.17 | 0.8766 | 0.51s | 16.11M | 7.850T |
| | LBAG (Li et al., 2023) | 34.91 | 0.9265 | 29.26 | 0.8788 | 0.51s | 16.11M | 7.852T |
| Transformers | Restormer (Zamir et al., 2022) | 34.85 | 0.9271 | 29.28 | 0.8785 | 3.72s | 26.13M | 6.741T |
| | Uformer-B (Wang et al., 2022) | 35.14 | 0.9277 | 29.92 | 0.8865 | 1.31s | 50.88M | 4.375T |
| | LMD-ViT | **35.42** | **0.9289** | **30.25** | **0.8938** | 0.56s | 54.50M | 1.485T |

Input  w/o pruning  w/ pruning  GT

Figure 5: Evaluations for sharp region quality preservation with or without the pruning strategy. Without pruning, the network modifies global pixels and blurs the sharp backgrounds. In contrast, with pruning, sharp backgrounds are preserved well.

and Transformer-based methods (Zamir et al., 2022; Wang et al., 2022) on the ReLoBlur dataset (Li et al., 2023) for local motion deblurring. As depicted in Figure 1 and Figure 3, LMD-ViT exhibits superior performance compared to other state-of-the-art methods, producing clearer outputs with enhanced details. Notably, the white stripes on the student's suit and the mobile phone show significant blur reduction without artifacts and closely resemble the ground truth. We notice that the number of preserved windows slightly exceeds the number of windows in the annotated blurry areas. This is to ensure that the preserved windows effectively cover as much of the blurry area as possible. Quantitative evaluation results are presented in Table 1, which bolds the best local deblurring performance. Compared with CNN-based methods, our proposed LMD-ViT achieves an improvement of 0.50 dB in PSNR and 0.95 dB in weighted PSNR, while maintaining a comparable or even faster inference speed. Compared with Transformer-based methods, LMD-ViT demonstrates significant reductions (-66%) in FLOPs and inference time without sacrificing performance (PSNR +0.28dB), thanks to our adaptive window pruning modules. The PSNR and SSIM scores of LMD-ViT also exceed other Transformer-based deblurring methods, because our proposed pruning strategy prohibits the network from destroying sharp regions. However, global deblurring methods (i.e., Uformer (Wang et al., 2022) and Restormer (Zamir et al., 2022)) treat every region equally and may distort or blur the sharp regions inevitably.

Additionally, we assess the global deblurring capabilities of LMD-ViT on the GoPro testing dataset (Nah et al., 2017) in Appendix C, and present more comparisons between our proposed method and baselines with local blur masks in Appendix D.

**User study on real-world photos.** To validate the effectiveness of our proposed model in real-world locally blurred images, we employ a static Sony industrial camera and a static Fuji XT20 SLR camera to capture 18 locally motion-blurred images, each with a resolution of 6000×4000 pixels. Subsequently, we conduct a comparative evaluation against the top 3 methods as listed in Table 1. As ground truths for these blurred images are unavailable, we organized a user study involving 30 participants with a keen interest in photography. Each participant is presented with randomly selected

Table 2: The effectiveness of window pruning. "$PSNR_w$", "$SSIM_w$", "Time", and "Params" denote weighted PSNR, weighted SSIM, inference time, and model parameters, respectively. When the pruning block is none, LMD-ViT does not prune and preserves all tokens.

| No. | Pruning block | $\beta$ | ↑PSNR | ↑SSIM | ↑$PSNR_w$ | ↑$SSIM_w$ | Time | Params | FLOPs |
|-----|---------------|---------|-------|-------|-----------|-----------|------|--------|-------|
| 1 | AdaWPT 1∼9 | 0.5 | 35.42 | 0.9289 | 30.25 | 0.8938 | 0.56s | 54.50M | 1.485T |
| 2 | AdaWPT 2∼8 | 0.5 | 35.41 | 0.9289 | 30.34 | 0.8928 | 0.70s | 53.94M | 1.592T |
| 3 | AdaWPT 3∼7 | 0.5 | 35.44 | 0.9290 | 30.38 | 0.8936 | 0.77s | 53.49M | 2.021T |
| 4 | AdaWPT 4∼6 | 0.5 | 35.37 | 0.9293 | 30.39 | 0.8931 | 1.07s | 53.06M | 3.105T |
| 5 | None | 0.5 | 35.36 | 0.9280 | 30.22 | 0.8930 | 1.30s | 50.39M | 4.376T |
| 6 | AdaWPT 1∼9 | 0.2 | 35.36 | 0.9289 | 30.21 | 0.8931 | 0.95s | 54.50M | 1.911T |
| 7 | AdaWPT 1∼9 | 0.3 | 35.41 | 0.9291 | 30.28 | 0.8934 | 0.80s | 54.50M | 1.671T |
| 8 | AdaWPT 1∼9 | 0.4 | 35.43 | 0.9290 | 30.28 | 0.8935 | 0.69s | 54.50M | 1.556T |
| 9 | AdaWPT 1∼9 | 0.6 | 35.35 | 0.9284 | 30.18 | 0.8922 | 0.49s | 54.50M | 1.405T |
| 10 | AdaWPT 1∼9 | 0.7 | 35.32 | 0.9281 | 30.14 | 0.8918 | 0.43s | 54.50M | 1.327T |

Table 3: The effectiveness of our blur mask annotation. "$PSNR_w$" and "$SSIM_w$" denote weighted PSNR, and weighted SSIM, respectively. Methods with "*" are trained with LBFMG's blur mask annotations [2]. Methods without "*" are trained with our blur mask annotations.

| No. | Methods | ↑PSNR | ↑SSIM | ↑$PSNR_w$ | ↑$SSIM_w$ |
|-----|---------|-------|-------|-----------|-----------|
| 1 | LBAG* (Li et al., 2023) | 34.83 | 0.9264 | 28.31 | 0.8711 |
| 2 | LBAG (Li et al., 2023) | 34.91 | 0.9265 | 29.26 | 0.8788 |
| 3 | LMD-ViT* | 35.31 | 0.9270 | 30.14 | 0.8911 |
| 4 | LMD-ViT | 35.42 | 0.9289 | 30.25 | 0.8938 |

deblurred images and tasked with selecting the most visually appealing one. The results, depicted in Figure 4, demonstrate that our proposed method exhibits robust performance on real-world locally motion-blurred images, showcasing sharper reconstructed edges and consistent content. Notably, it emerges as the preferred choice among participants when compared to alternative approaches. Further details of the user study are provided in Appendix F.2.

## 3.3 ANALYSES

To examine the effectiveness of our proposed method, we conduct separate analyses focusing on the window pruning strategy and blur mask annotations in this section. Due to space constraints, details of pruning accuracy, and analyses of W-LeFF, the number of feature channels, joint training strategy, input resolutions, and special cases are presented in Appendix E.

### 3.3.1 WINDOW PRUNING STRATEGY

We analyze the effects of our window pruning strategy in three aspects: the number of pruning blocks, the pruning threshold $\beta$ during inference, and pruning accuracy.

**The number of pruning blocks.** We fix the pruning threshold $\beta = 0.5$ and change the number of pruning blocks. For blocks without pruning, we force the decision maps $\mathbf{D}$ in Equation 4 to be the all-one matrix. From line 1 to line 5 of Table 2, we find that pruning more blocks results in fewer model parameters, a higher inference speed, and more dropping scores. Notably, although we apply a window pruning mechanism in all 9 blocks, the scores outperform other baseline models listed in Table 1. Additionally, we conduct visual comparisons of all-pruned architecture (line 1) with non-pruning architecture (line 5), as shown in Figure 5. In the all-pruned network, most of the sharp regions are pruned and are not destroyed. In contrast, the sharp regions turn blurry or distorted when processed by the non-pruned network. This indicates that our adaptive window pruning mechanism can prevent the sharp regions from being destroyed. The non-pruning network treats every region equally like global deblurring networks (e.g., MIMO-UNet, (Cho et al., 2021), Uformer (Wang et al., 2022) and Restormer (Zamir et al., 2022)) that may harm the sharp regions inadvertently.

**Pruning threshold $\beta$.** We fix the pruning blocks and adjust the pruning threshold $\beta$ from 0.2 to 0.7 with 0.1 as the interval. Comparing line 1, lines 6 to line 10 in Table 2, we find that the testing

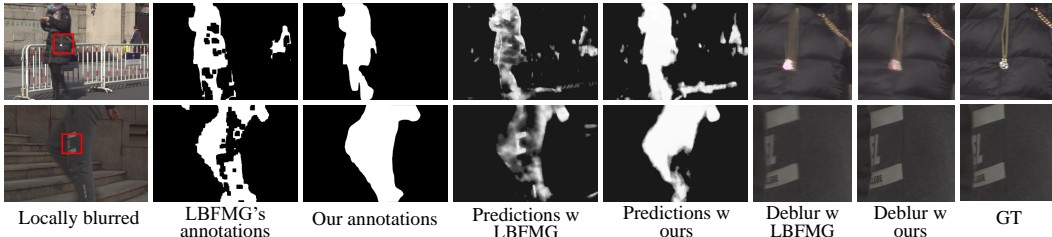

| Locally blurred | LBFMG's annotations | Our annotations | Predictions w LBFMG | Predictions w ours | Deblur w LBFMG | Deblur w ours | GT |

Figure 6: Visualizations of LMD-ViT's blurriness confidence map trained with different kinds of annotations. LMD-ViT predicts better blurriness confidence maps with our annotation than those produced with the annotations provided by LBFMG [2] (Li et al., 2023).

performance slightly varies with different pruning thresholds $\beta$, resulting in different confidence levels and decision boundaries. Inferring with a lower nor a higher pruning threshold is neither reasonable, because the former makes the confidence predictor more inclusive, which potentially leads to fewer sharp windows to be pruned and a slower inference speed, while the latter makes the confidence predictor more conservative, which leads to faster inference speed but filters out the necessary blurry windows. To achieve a balance, we choose $\beta = 0.5$ as it obtains relatively high evaluation scores and fast inference speed.

**Pruning accuracy** Our proposed AdaWPT enables blur region selection as well as removing sharp windows. From the first pruning block (AdaWPT 1) to the last pruning block (AdaWPT 9), the preserved patches gradually correspond to the ground-truth blur annotations, as shown in Figure 1. To assess the local motion deblurring capability of our proposed LMD-ViT, we measure the mask prediction accuracy on the ReLoBlur test dataset using the formula: $accuracy = \frac{TP+TN}{N}$, where $TP$ is the count of correctly identified blurry pixels, $TN$ represents accurately identified sharp pixels, and $N$ stands for the total pixel count. Our model achieves a peak accuracy of 94.51% among various AdaWPT blocks, which predominantly reflects an effective recognition of local motion blurs. Notably, the accuracy rises from AdaWPT 1 to 8, while AdaWPT 9's accuracy dips possibly due to its skip connection with AdaWPT 1. We detail the accuracies of all blocks in Appendix E.5.

### 3.3.2 BLUR MASK ANNOTATION

To verify the effectiveness of our blur mask annotations, we conduct experiments on a local deblurring method LBAG (Li et al., 2023) and our proposed LMD-ViT, as illustrated in Table 3. Comparing line 1 and line 2, we observe that our manually annotated blur masks enhance the evaluation scores. This observation is further supported by the comparison between line 3 and line 4. We provide visualizations of mask predictions and deblurred results of LMD-ViT in Figure 6. It is evident that LBFMG's[2] masks lead to more recognition errors of local blurs. This is due to the presence of holes and the noise in LBFMG's[2] masks (Li et al., 2023), which may confuse AdaWTP in predicting masks and selecting blurry tokens. However, our smooth and continuous annotations greatly improve the performance of blur mask predictions, thus aiding LMD-ViT in presenting more promising deblurring results.

## 4 CONCLUSION

We have presented an adaptive and efficient approach, LMD-ViT, a sparse vision Transformer for restoring images affected by local motion blurs. LMD-ViT is built upon our novel adaptive window pruning Transformer blocks (AdaWPT), which employ blur-aware confidence predictors to estimate the level of blur confidence in the feature domain. This information is then used to adaptively prune unnecessary windows in low-confidence regions. To train the confidence predictor, we designed an end-to-end reconstruction loss with Gumbel-Softmax re-parameterization, along with a pruning loss guided by our carefully annotated blur masks. Extensive experiments demonstrate that our method effectively eliminates local motion blur while ensuring minimal deformation of sharp regions, resulting in a significant improvement in image quality and inference speed. Due to limited page space, we discuss the limitations and broader impacts in Appendix G.

---

[1]LBFMG is an automatic blur mask annotation method proposed by Li et al. (2023)

ACKNOWLEDGMENTS

This study is supported under the RIE2020 Industry Alignment Fund Industry Collaboration Projects (IAF-ICP) Funding Initiative, as well as cash and in-kind contributions from the industry partner(s). This study is also funded by the China Scholarship Council.

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

# A  RELATED WORK

**Single image deep motion deblurring.** The task of deep motion deblurring for single images originated from global deblurring (Zhang et al., 2018; Ren et al., 2021; Tao et al., 2018; Zamir et al., 2021; Zhang et al., 2021; Zhou et al., 2022). Pioneering deep global motion deblurring works utilize CNNs as basic layers and achieve promising improvements in image quality. Among them, Deep-Deblur (Nah et al., 2017), a multi-scale convolutional neural network, performs residual blocks to increase convergence speed. DeblurGAN (Kupyn et al., 2018) and DeblurGAN-v2 (Kupyn et al., 2019) introduce GANs and a perceptual loss to improve subjective quality. HINet (Chen et al., 2021) applies Instance Normalization to boost performance. Recently, a CNN-based local motion deblurring method, LBAG (Li et al., 2023), bridges the gap between global and local motion deblurring by inserting gate modules at the end of MIMO-UNet architecture (Cho et al., 2021). It predicts differentiable blur masks to reduce sharp backgrounds from modifications and guide the network to deblur locally. There are also CNN-based methods that involve spatially variant predictions for blind image reconstruction (Liang et al., 2021b). Although the performance is significantly improved, CNN-based methods suffer from the content-independent interactions between images and convolution kernels, as well as the limitations of long-range dependency modeling.

Given the Vision Transformer's (ViT) (Dosovitskiy et al., 2021) ability to capture long-range dependencies, its application to global deblurring tasks has seen a surge of interest. For example, Uformer (Wang et al., 2022) employs window-based self-attention with a learnable multi-scale restoration modulator to capture both local and global dependencies. Restormer (Zamir et al., 2022) utilizes multi-head attention and a feed-forward network to achieve long-range pixel interactions. In this paper, we build a Transformer-based local motion deblurring framework, LMD-ViT, that adaptively selects windows relevant to blurry regions for window-based self-attention and feed-forward operations, simultaneously benefiting from long-range modeling.

**Vision Transformer acceleration.** Transformers have proven valuable in deblurring tasks, yet their direct application in local motion deblurring for high-resolution images presents challenges concerning computational efficiency. To solve the heavy computation problem of global self-attention in Transformers, researchers have presented several techniques. For example, Wang et al. adopted pyramid structures and spatial-reduction attention (Wang et al., 2021) in image classification, object detection, and segmentation tasks. Some methods partition image features into different windows and perform self-attention on local windows (Vaswani et al., 2021; Yang et al., 2021; Wang et al., 2022) for image restoration tasks. Some image classification methods gradually reduce tokens in processing by token-halting (Meng et al., 2022; Rao et al., 2021; Yin et al., 2022) or token-merging (Liang et al., 2022; Bolya et al., 2022). Inspired by these advancements, we develop adaptive window pruning blocks (AdaWPT) to eliminate unnecessary tokens and focus deblurring only on blurred regions, which improves image quality and enables inference speed-up without compromising sharp regions.

# B  LMD-VIT ARCHITECTURE AND MODEL HYPER-PARAMETERS

As shown in Figure 2 of our main paper, our proposed LMD-ViT is a U-shaped network with one in-projection layer, four encoder stages, one bottleneck stage, four decoder stages, and one out-projection layer. Skip connections are set up between the encoder stage and the decoder stage. A locally blurred input image $\mathbf{B} \in \mathbb{R}$ with a shape $H \times W \times 3$ firstly goes through an in-projection block, which consists of a $3 \times 3$ convolutional layer, a LeakyReLU layer, and a layer normalization block, to extract low-level features as a feature map $\mathbf{X} \in \mathbb{R}$. The feature map then passes four encoder stages, each of which includes a series of AdaWPT Transformer blocks and one down-sampling layer. AdaWPT uses a blur-aware confidence predictor and Gumble-Softmax re-parameterization to select blur-related tokens. Only the selected tokens are forwarded to Transformer layers including window-based self-attention (W-MSA), window-based locally-enhanced feed-forward layer (W-LeFF), and layer normalization (LN). The down-sampling layer down-samples the feature map size by 2 times and doubles the channels using $4 \times 4$ convolution with stride 2. The feature map's shape turns to $\frac{H}{2^i} \times \frac{W}{2^i} \times 3, i \in \{1, 2, 3, 4\}$ after $i$ encoder stages, and has the smallest resolution in the bottleneck, where it can sense the longest dependencies in two AdaWPTs. After the bottleneck stage, the feature map goes through four decoder stages, each of which owns an up-sampling layer and a series of AdaWTP blocks. The up-sampling layer uses $2 \times 2$ transposed convolution with

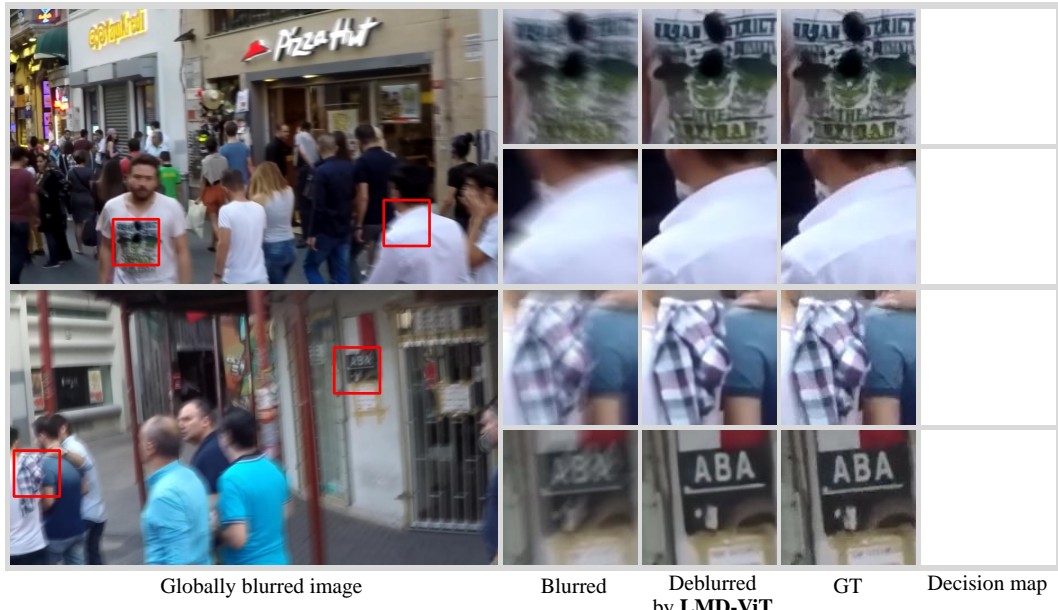

|  Globally blurred image | Blurred | Deblurred by **LMD-ViT** | GT | Decision map |

Figure 7: Global deblurring performance of our proposed LMD-ViT. The decision map denotes the pruning decisions. With a globally blurry input, the predictedblurriness confidenceapproaches 100%. Therefore, our network preserves all windows and nearly all the values in the decision map are equal to 1, which is represented as an all-white color.

Table 4: Quantitative comparisons on the global deblurring dataset. All the methods are trained with the ReLoBlur dataset (Li et al., 2023) and the GoPro dataset (Nah et al., 2017) together.

| Evaluation metrics | Restormer (Zamir et al., 2022) | Uformer-B (Wang et al., 2022) | LMD-ViT (ours) |
|---|---|---|---|
| PNSR | 32.15 | 32.51 | 32.16 |
| SSIM | 0.9305 | 0.9376 | 0.9318 |

stride 2 to reduce half of the feature channels and double the size of the feature map. The features put into the AdaWPT blocks are concatenations of the up-sampled features and the corresponding features from the symmetrical encoder stages through skip connections. Finally, the feature map passes the out-projection block which reshapes the flattened features to 2D feature maps and applies a $3 \times 3$ convolution layer to obtain a residual image $\mathbf{R}$. The restored sharp image $\mathbf{S}'$ is obtained by $\mathbf{S}' = \mathbf{B} + \mathbf{R}$.

## C  GLOBAL DEBLURRING PERFORMANCES

Our proposed LMD-ViT integrates both local and global motion deblurring. To demonstrate its capability in global deblurring, we evaluate our model on the GoPro testing dataset (Nah et al., 2017). With a globally blurry input, the predicted blurriness confidence approaches 100%. Consequently, the decision layer of LMD-ViT retains all windows, depicted in white in the $5^{th}$ column of Figure 7. We also compare our proposed models with other state-of-the-art Transformer-based deblurring methods in Table 4. The scores of our method slightly lag behind Uformer (Wang et al., 2022) due to the utilization of a window pruning strategy in LMD-ViT, which does not achieve 100% accuracy. As a result, some tokens may miss deblurring manipulations in certain blocks, leading to less clear outputs. The visual comparison on the GoPro testing dataset (Nah et al., 2017) shows that LMD-ViT obtains comparable performance to the state-of-the-art global deblurring Transformers, as shown in Figure 8. Our proposed method can effectively eliminate global blurriness and restore sharp textures, showcasing its competitiveness against other baseline methods.

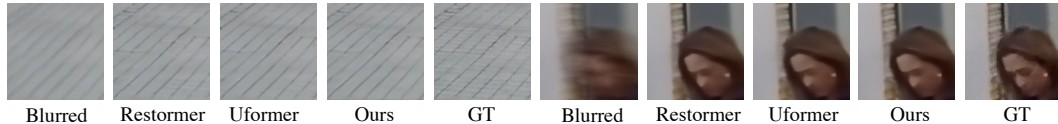

Figure 8: Visual results of global motion deblurring.

Table 5: Quantitative comparisons between LMD-ViT and baselines with local blur masks. "PSNR$_w$", and "SSIM$_w$" denote weighted PSNR, and weighted SSIM, respectively. We bold the best result under each evaluation metric.

| Methods | ↑PSNR | ↑SSIM | ↑PSNR$_w$ | ↑SSIM$_w$ | Inference Time |
|---|---|---|---|---|---|
| Restormer w/ mask | 34.88 | 0.9259 | 29.44 | 0.88055 | 3.75s |
| Uformer-B w/ mask | 35.23 | 0.9277 | 30.22 | 0.8935 | 1.33s |
| LMD-ViT (ours) | **35.42** | **0.9285** | **30.25** | **0.8938** | **0.56s** |

# D COMPARISONS BETWEEN LMD-ViT AND BASELINES WITH LOCAL BLUR MASKS

In Table 1, both LBAG (Li et al., 2023) and our proposed LMD-ViT are trained utilizing local blur masks, whereas the baseline models, including Restormer (Zamir et al., 2022) and Uformer(Wang et al., 2022), are not originally designed with this feature. To ensure a fair comparison, we retrain the top three baselines mentioned in Table 1 with local blur masks. Given that these baselines' architectures lack a dedicated mask input, we integrate the gated block from LBAG (Li et al., 2023), which is a local blur-aware technique. Moreover, we apply our method's reconstruction loss (as detailed in Equation 9) to Uformer (Wang et al., 2022) and Restormer (Zamir et al., 2022). The results in Table 5 reveal that upon training with local blur masks, Uformer and Restormer indeed witness improvements in their PSNR and SSIM scores. Despite this enhancement, their performance still trails behind ours, and they require a longer inference time. This substantiates the efficacy of our backbone and the adopted window pruning strategy.

# E FURTHER DISCUSSION OF ABLATION STUDY

## E.1 THE EFFECTIVENESS OF W-LeFF

To save computational costs as well as inference time, we apply a window-based local-enhanced feed-forward layer (W-LeFF) instead of a locally-enhanced feed-forward layer (LeFF) (Wang et al., 2022). In W-LeFF, only the non-pruned windows are processed by a feed-forward mechanism. To ensure a fair comparison between W-LeFF and LeFF, we evaluate them on the all-pruning LMD-ViT network architecture separately. As shown in Table 6, we observe that W-LeFF achieves nearly identical performance to LeFF. Hence, substituting LeFF with W-LeFF does not compromise the local motion deblurring capabilities while simultaneously accelerating the inference speed.

## E.2 THE EFFECT OF FEATURE CHANNELS

The number of feature channels also affects the ability of neural networks. With a larger number of feature channels, a neural network can capture more intricate and complex relationships in the input data, resulting in better performance. To verify the capability of our proposed network architecture, we train our network with dimensions 16 and 32 respectively, and compare it with CNN-based LBAG (Li et al., 2023) with aligned model parameters, as shown in Table 7. The comparison between line 3 and line 4 shows an improvement with increased feature channels in LMD-ViT because a larger number of feature channels provides a larger number of meaningful features or attributes, which can be beneficial for window selection and feature manipulation. The comparison between line 2 and line 4 implies that, under approximate model parameters, our proposed model with the adaptive window pruning mechanism is more suitable for the local motion deblurring task with better evaluation scores, fewer model parameters, and faster inference speed.

Table 6: The effectiveness of W-LeFF in the proposed LMD-ViT. "PSNR$_w$", "SSIM$_w$", "Time", "Params", and "FLOPs" denote weighted PSNR, weighted SSIM, inference time, model parameters, and model complexity, respectively.

| No. | Methods | Pruning block | ↑PSNR | ↑SSIM | ↑PSNR$_w$ | ↑SSIM$_w$ | Time | Params | FLOPs |
|-----|---------|---------------|-------|-------|-----------|-----------|------|--------|-------|
| 1 | LMD-ViT w **LeFF** | AdaWPT 1~9 | 35.45 | 0.9288 | 30.24 | 0.8935 | 0.86s | 54.50M | 1.485T |
| 2 | LMD-ViT w **W-LeFF** | AdaWPT 1~9 | 35.42 | 0.9285 | 30.25 | 0.8938 | 0.56s | 54.50M | 1.485T |

Table 7: The effect of feature dimension in the proposed LMD-ViT. "Feature channels", "PSNR$_w$", "SSIM$_w$", "Time", "Params", and "FLOPs" denote the feature channels of each block, weighted PSNR, weighted SSIM, inference time, model parameters, and model complexity respectively.

| No. | Methods | Feature channels | ↑PSNR | ↑SSIM | ↑PSNR$_w$ | ↑SSIM$_w$ | Time | Params | FLOPs |
|-----|---------|------------------|-------|-------|-----------|-----------|------|--------|-------|
| 1 | LBAG (Li et al., 2023) | 32-64-128 | 34.91 | 0.9265 | 29.26 | 0.8788 | 0.51s | 16.11M | 7.852T |
| 2 | LBAG-Large (Li et al., 2023) | 60-120-240 | 34.93 | 0.9266 | 29.36 | 0.8871 | 1.13s | 56.58M | 17.659T |
| 3 | LMD-ViT-Small | 16-32-64-128-256 | 34.98 | 0.9259 | 29.89 | 0.8907 | 0.23s | 21.59M | 0.311T |
| 4 | LMD-ViT | 32-64-128-256-512 | 35.42 | 0.9285 | 30.25 | 0.8938 | 0.56s | 54.50M | 1.485T |

### E.3 THE EFFECTIVENESS OF THE JOINT TRAINING STRATEGY

We train our proposed LMD-ViT and baseline methods with the ReLoBlur dataset (Li et al., 2023) and the GoPro dataset together mainly for two reasons. Firstly, joint training could prevent our model from over-fitting and improve the model's robustness. Secondly, we expect our proposed LMD-ViT both deblur globally and locally. Training with the ReLoBlur dataset (Li et al., 2023) and the GoPro dataset (Nah et al., 2017) together improves both the local and global deblurring performances, as shown in Table 8.

### E.4 LOCAL DEBLURRING RESULTS UNDER DIFFERENT RESOLUTIONS

We enrich our findings by including results for various resolutions. Employing the nearest interpolation, we down-sample the ReLoBlur testing data and conduct comparisons between our proposed model and the top 2 baseline methods in Table 9. We find that the local deblurring results of our method and the baselines drop. However, our proposed method outweighs baseline methods in terms of PSNR and SSIM scores. Moreover, it achieves fast inference in low-resolution images.

### E.5 MASK PREDICTION ACCURACY, PRECISION AND RECALL

To further evaluate the local motion deblurring ability of our proposed LMD-ViT, we gauge the mask prediction accuracy, precision and recall of our model on the ReLoBlur testing dataset using the formulas:

$$
\begin{aligned}
accuracy &= \frac{TP + TN}{N}, \\
precision &= \frac{TP}{TP + FP}, \\
recall &= \frac{TP}{TP + FN},
\end{aligned}
\tag{10}
$$

where $TP$ denotes the number of pixels correctly predicted as blurry, $TN$ signifies the count of pixels accurately predicted as sharp, $FN$ denotes the count of blurry pixels predicted as sharp, and $N$ represents the total pixel count. The mask prediction accuracies, precisions and recalls exhibit variability across different AdaWPT blocks, as shown in Figure 9. Most AdaWPT blocks exhibit high prediction accuracies, indicating the effective recognition of local blurs by our proposed model. The first two learnable blocks show comparatively lower accuracies and precisions. The accuracy and precision of AdaWPT 9 drops, potentially influenced by its skip connection with AdaWPT 1.

Table 8: The effectiveness of joint training with the ReLoBlur dataset (Li et al., 2023) and the GoPro dataset (Nah et al., 2017). We evaluate in terms of the PSNR and SSIM metrics.

| No. | Training data | | Training mask | | Testing data | |
|---|---|---|---|---|---|---|
| | ReLoBlur | ReLoBlur + GoPro | LBFMG's mask[2] | Our mask | ReLoBlur | GoPro |
| 1 | ✓ | | ✓ | | 35.24 dB / 0.9283 | 31.60 dB / 0.9252 |
| 2 | ✓ | | | ✓ | 35.29 dB / 0.9280 | 31.74 dB / 0.9271 |
| 3 | | ✓ | ✓ | | 35.31 dB / 0.9270 | 32.14 dB / 0.9217 |
| 4 | | ✓ | | ✓ | 35.42 dB / 0.9285 | 32.16 dB / 0.9318 |

Table 9: Comparisons for local motion deblurring under different resolutions. We test our proposed LMD-ViT with baseline methods with down-sampled ReLoBlur testing data. LMD-ViT significantly outweighs other methods and achieves efficient inference. "$PSNR_w$", "$SSIM_w$", and "Time" denote weighted PSNR, weighted SSIM, and inference time respectively. "*" denotes the down-sampled ReLoBlur testing dataset.

| Image Resolution | Methods | ↑PSNR | ↑SSIM | ↑$PSNR_w$ | ↑$SSIM_w$ | Time | FLOPs |
|---|---|---|---|---|---|---|---|
| ReLoBlur* | Restormer (Zamir et al., 2022) | 34.28 | 0.9130 | 29.01 | 0.8510 | 0.99s | 1.723T |
| (Li et al., 2023) | Uformer (Wang et al., 2022) | 33.89 | 0.9053 | 28.65 | 0.8424 | 0.36s | 1.158T |
| ($1076 \times 718$) | LMD-ViT (ours) | 34.59 | 0.9176 | 29.39 | 0.8586 | 0.23s | 266.074G |
| ReLoBlur* | Restormer (Zamir et al., 2022) | 33.10 | 0.8903 | 28.53 | 0.8376 | 0.26s | 450.210G |
| (Li et al., 2023) | Uformer(Wang et al., 2022) | 32.88 | 0.8747 | 28.38 | 0.8293 | 0.25s | 321.657G |
| ($538 \times 359$) | LMD-ViT (ours) | 33.52 | 0.8978 | 28.85 | 0.8451 | 0.10s | 162.627G |

### E.6 RESULTS OF SPECIAL CASES

We subjected our proposed model to testing using two distinctive input scenarios. In the case of globally blurred images sourced from the GoPro testing dataset, the corresponding masks exhibit uniformly high values, and the decision maps manifest predominantly in an all-white color, as depicted in Appendix C and visually represented in Figure 7. Conversely, when inputting sharp images from the GoPro testing dataset, the mask values are uniformly low, resulting in decision maps that predominantly display an all-black color, as illustrated in Figure 10. The results of these extreme experiments demonstrate the model's ability to effectively distinguish between blurred and sharp images, showcasing its robustness.

## F MORE VISUAL RESULTS OF LOCAL MOTION DEBLURRING

### F.1 MORE VISUAL COMPARISONS BETWEEN LMD-VIT AND THE COMPARED METHODS

We provide more visual results of our proposed LMD-ViT and other baseline methods for local motion deblurring, as shown in Figure 11 and Figure 12. The performance of LMD-ViT surpasses that of other state-of-the-art methods, yielding output images with improved clarity and enhanced details.

### F.2 MORE VISUAL RESULTS OF USER STUDY

We provide more visual results for real-world local motion deblurring, which are used for conducting user study, as shown in Figure 13. We conduct a comparison between our method and the top 3 methods listed in Table 1 of the main paper. The visual results show that our proposed method exhibits robust deblurring capability and outperforms the state-of-the-art methods on real-world locally blurred images.

## G LIMITATIONS

Like other local motion deblurring methods (Nah et al., 2017; Kupyn et al., 2018; 2019; Chen et al., 2021; Cho et al., 2021; Li et al., 2023; Wang et al., 2022; Zamir et al., 2022), our proposed LMD-

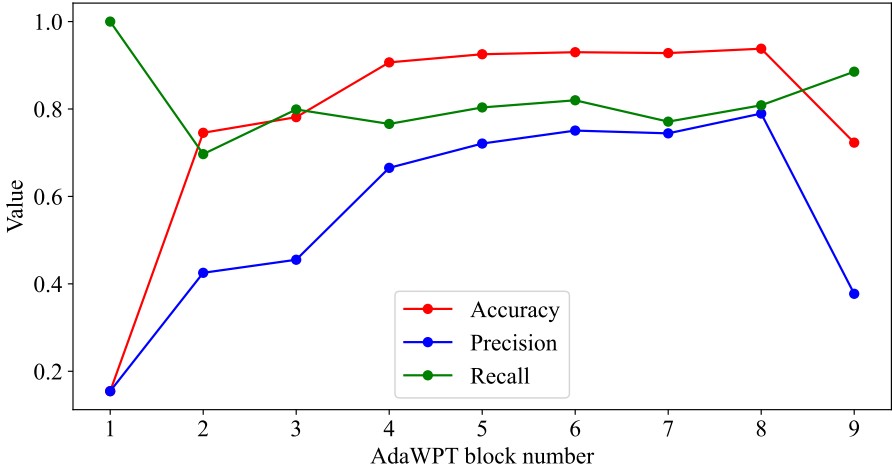

Figure 9: The mask prediction accuracy, precision, and recall on the ReLoBlur testing dataset.

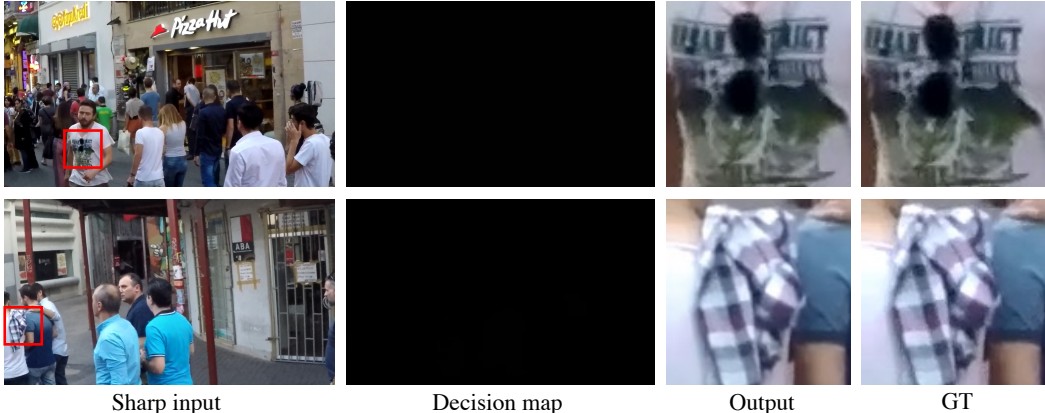

Figure 10: Results of special cases. When inputting static sharp images, the decision maps tend to be all-zero metrics and colored in black. The output is very consistent with the ground truth because LMD-ViT prohibits nearly all tokens from going through Transformer layers.

ViT is not capable of real-time deblurring. However, we plan to optimize our structure so that it can be applied in real time.

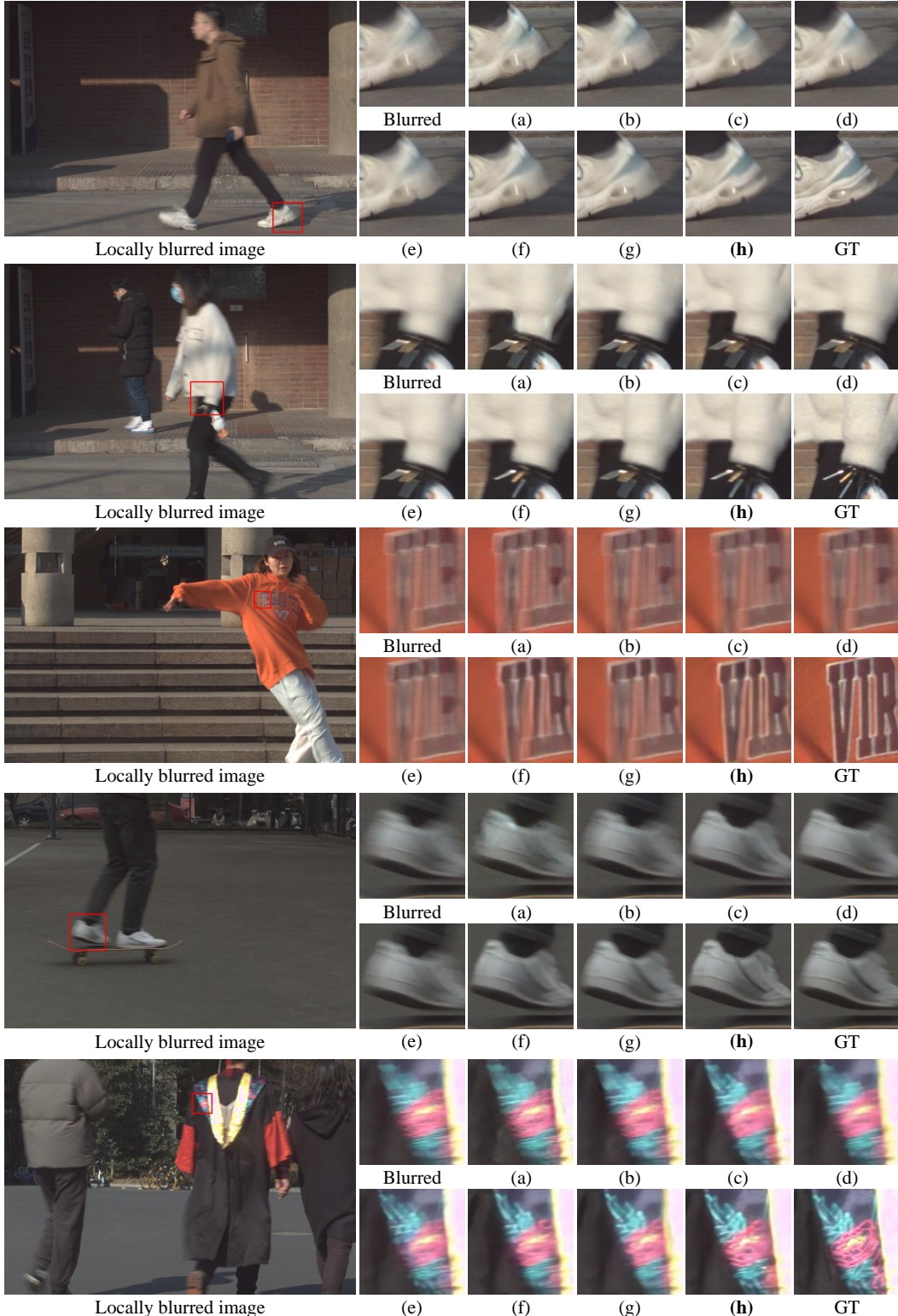

Figure 11: Visual comparing results of state-of-the-art methods for local motion deblurring. (a) DeepDeblur (Nah et al., 2017); (b) DeblurGAN_v2 (Kupyn et al., 2019); (c) HINet (Chen et al., 2021); (d) MIMO-UNet (Cho et al., 2021); (e) LBAG (Li et al., 2023); (f) Restormer (Zamir et al., 2022); (g) Uformer (Wang et al., 2022); (h) LMD-ViT (ours).

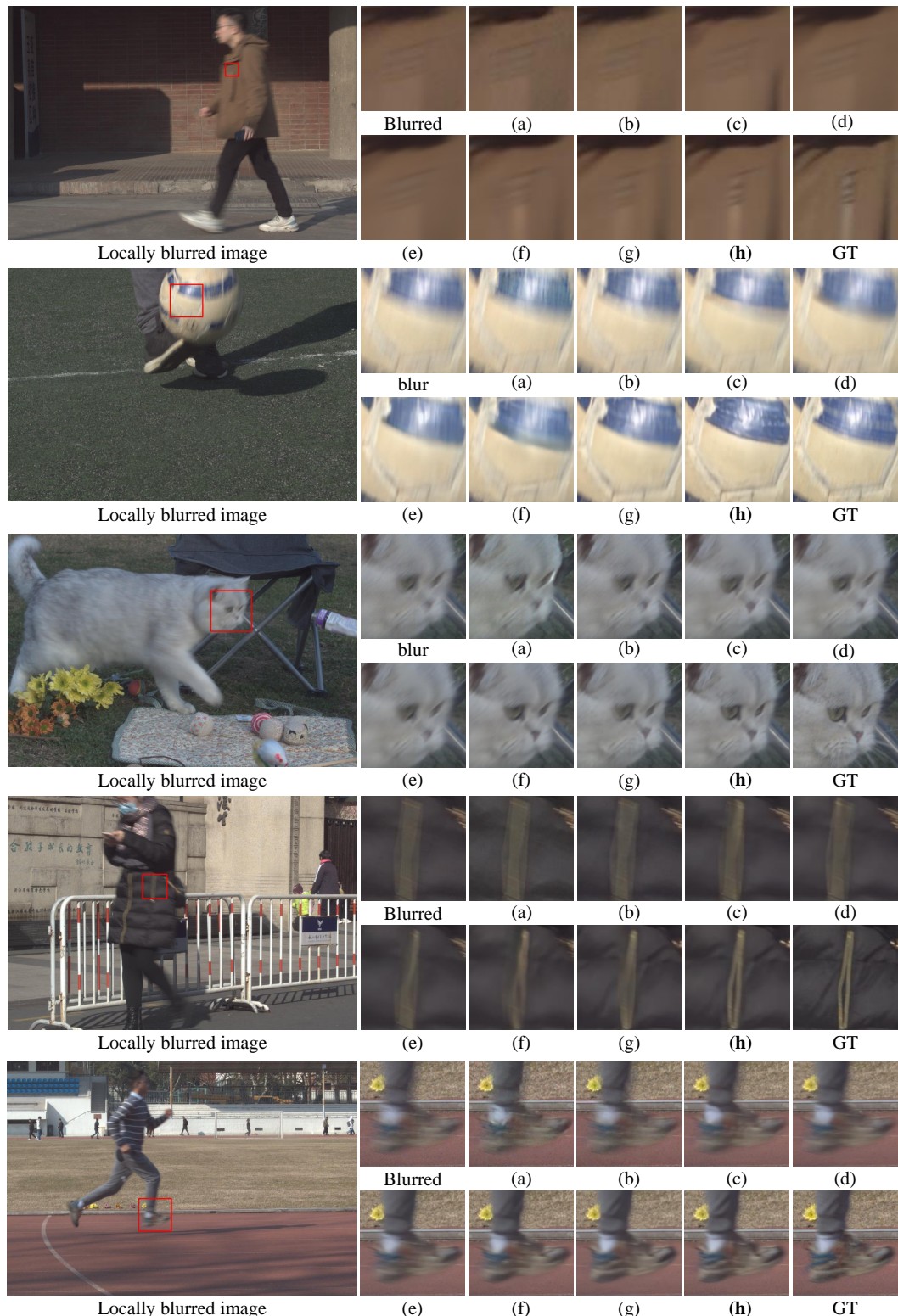

Figure 12: More visual comparing results of state-of-the-art methods for local motion deblurring. The annotations are the same as that in Figure 11.

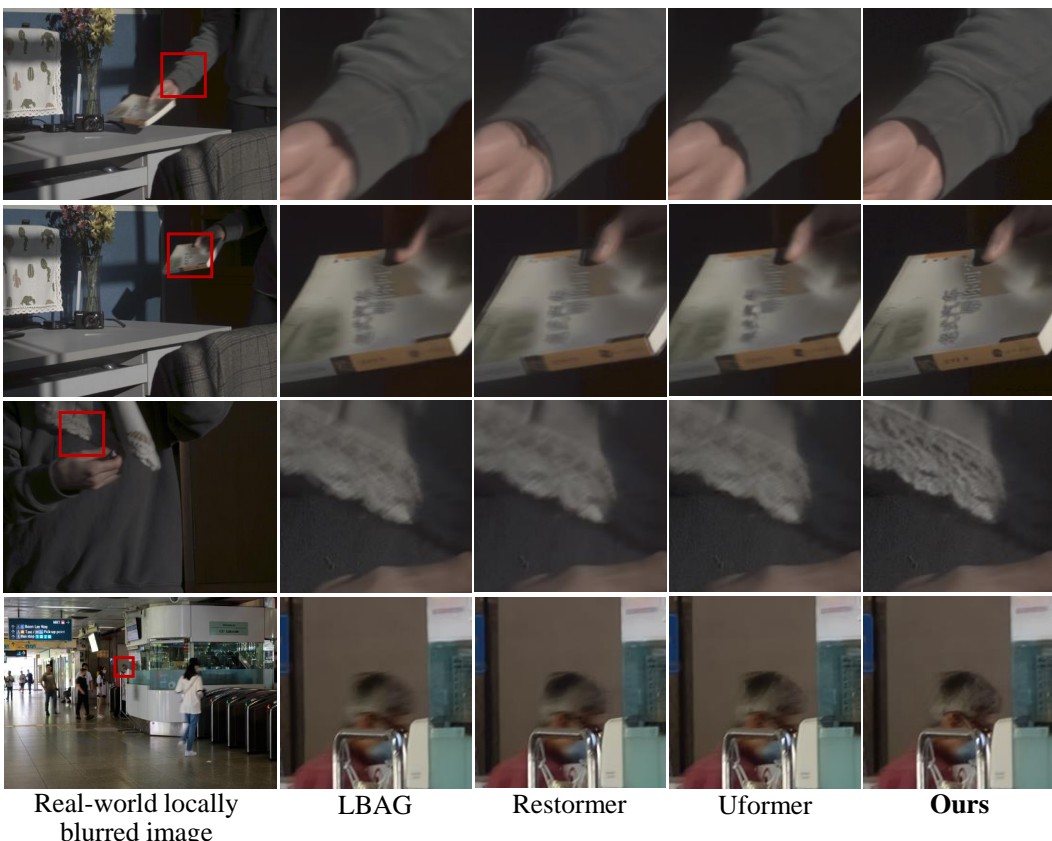

|  |  |  |  |  |
| --- | --- | --- | --- | --- |
| Real-world locally blurred image | LBAG | Restormer | Uformer | **Ours** |

Figure 13: Visual results of user study for real-world local motion deblurring. We compare our proposed LMD-ViT with the top 3 methods, i.e., LBAG (Li et al., 2023), Restormer (Zamir et al., 2022), and Uformer (Wang et al., 2022), listed in Table 1 of the main paper.

