# OpenReview forum: "Adaptive Window Pruning for Efficient Local Motion Deblurring"
_ICLR.cc/2024/Conference — ICLR 2024 poster_

### Official Review · Reviewer_S2zK · 2023-10-15

**Soundness:** 3 good
**Presentation:** 4 excellent
**Contribution:** 3 good
**Rating:** 8
**Confidence:** 5

**Summary:**

This paper proposes a local motion deblurring Transformer with adaptive window pruning, which only deblur the active (blurry) windows. The windows are classified as active/inactive according to the predicted bluriness confidence score.

**Strengths:**

Overall, I think this paper has a novel idea and achieves good results in terms of performance and efficiency. I tend to accept this paper due to following reasons.

1. adaptive window pruning that saves computation on unnecessary attention windows.

2. bluriness confidence prediction that works well for both local motion prediction and global motion prediction

3. annotated local blur masks on ReLoBlur

4. well-designed experiments, well-presented figures and well-written paper.

**Weaknesses:**

Below are some concerns and suggestions.

1. Since the confidence predictor only uses MLP layers. How many pixels did you shift? Is the feature shift necessary to enlarge the receptive field of the neighbourhood?

2. What is the mask prediction accuracy on validation set?

3. How did you decide the border when annotating the masks for blurry moving objects?

4. If a patch is always abandoned, how is it processed? What layers it will be passed into during inference?

5. It could be better to provide the results of two special cases (masks are all-ones/all-zeros) in the tables as a reference.

6. Why did you only report real-world results on large images? Do you have a chart for comparison on PSNR/FLOPs/runtime under different image resolutions.

7. The key of this method is the adaptive window pruning. It is better to provide an ablation study with the rest tricks as a baseline (i.e., no window pruning in training and testing).


Minor:

1. Unfinished paragraph in page 2.

2. For figure 1, I think it might be better to visualise the attention window borders, for example, by adding solid lines to show the windows.

3. The summarised contributions are a bit overlapped (point 1 and point 2). I think it's better to claim adaptive window pruning and bluriness confidence prediction as two contributions.

4. I think there is no need to distinguish between AdaWPT-F and AdaWPT-P. Just be simple and united. The name of AdaWPT is enough.

5. Comparison in Figure 6 is not visible.

6. Is there an example on a globally clear image (e.g., a still scene). Contrary to Figure 7, will the decision map be all zeros?

7. Are there other similar works in image restoration/deblurring? Are there some connections between this method and some "blur kernel prediction + restoration" methods (e.g. Mutual Affine Network for Spatially Variant Kernel Estimation in Blind Image Super-Resolution)?

**Questions:**

See weakness.

---

> ### Author Response · Authors · 2023-11-19
> **Responses to questions 1-6**
>
> **The number of shifting pixels and the necessity.** In each AdaWPT block, we shift 4 tokens, which is half of the window size. The number of shifting pixels is proportional to the token size of each block. For example, for AdaWPT 1 and 9, whose resolution is $512\times512$, the number of shifting pixels is $4\times4$. Similarly, for AdaWPT 2 and 8, the number of shifting pixels is $8\times8$... For AdaWPT 5, the number of shifting pixels is $64\times64$. As the block's resolution decreases, the number of shifting pixels increases, effectively expanding the receptive field. This shifting operation aids in enlarging the receptive field of the local neighborhood, thereby enhancing the deblurring performance. Noticeably, we have trained our proposed network without the Feature Shift/Reverse layer, and the absence of these layers results in a PSNR drop of approximately 0.1dB.
>
> **The prediction accuracy on the validation set.** We calculate the prediction accuracy by $(TP + TN) / N$, where $TP$ refers to the number of pixels that our model correctly predicts to be blurry, $TN$ refers to the number of pixels our model correctly predicts to be sharp, and $N$ is the total number of pixels. The mask prediction accuracies vary in different AdaWPT blocks. The highest is 94.51\%. Figure 12 in the new version of our Appendix reports the accuracy of each AdaWPT block. Most AdaWPT blocks exhibit high prediction accuracies, indicating the effective recognition of local blurs by our proposed model. We have incorporated this discussion into the new version of Appendix E.6.
>
> **About the blur mask borders.** We subtract the sharp and blurred images to observe the differences in local regions. This difference map helps us to determine the approximate position of the blurriness and the border. To prevent leakage, we extend the blurry region by 5 pixels, ensuring that all pixels within the annotated mask are considered blurry. Regarding pixels between blurry and sharp areas, we adhere to a principle: a diffuse spot encompassing more than 5 pixels is categorized as a blurry region, while conversely, it is classified as a sharp region. We add this discussion in the new version of Appendix C.
>
> **How abandoned patches are processed.** Because the abandoned patches are sharp, they do not go through Transformer layers. They directly go to the Window Compound layer during inference. The Window Compound layer is incorporated \textcolor{red}{to integrate the abandoned windows and selected windows into a unified feature map in inference. This is explained in the new version of Section 2.2, highlighted in red.
>
> **Results of special cases (masks are all-ones/all-zeros).** I guess you refer to the results of inputting globally blurred images and sharp images. When inputting globally blurred images, the masks are all one, and the decision maps are nearly in all-white color, as is shown in Appendix D and Figure 7. When inputting globally blurred images, the masks are all zero, and the decision maps are nearly in all-black color. The results of these extreme experiments demonstrate the model’s ability to effectively distinguish between blurred and sharp
> images, showcasing its robustness. We add the results and discussion in the new version of Appendix E.7, highlighted in red.
>
> **Results under different image resolutions.** In the initial version of our paper, we presented results based on large images due to the ReLoBlur testing data exclusively comprising images with substantial resolutions. In this revision, we augment our findings by including results for various resolutions in the subsequent chart. Employing the nearest interpolation, we down-sample the ReLoBlur testing data and conduct comparisons between our proposed model and baselines using both middle-resolution and small-resolution images. The results depicted in the following chart demonstrate that our proposed LMD-ViT outperforms other methods across all resolutions, delivering rapid and efficient inference. This discussion has been incorporated into the updated version of Appendix E.4.
>
> | Image Resolution | Methods  | $\uparrow$PSNR | $\uparrow$SSIM  | $\uparrow$PSNR$_w$ | $\uparrow$SSIM$_w$ | Inference time | FLOPs |
> |----------|----------|----------|----------|----------|----------|----------|----------|
> | ReLoBlur* ($1076\times718$) | Restormer [1] | 34.28 | 0.9130  | 29.01 | 0.8510 | 0.99s | 1.723T |
> | ReLoBlur* ($1076\times718$)   | Uformer [2] | 32.88 | 0.8747  | 28.38 | 0.8293 | 0.25s | 1.158T|
> | ReLoBlur* ($1076\times718$)   | LMD-ViT (ours) | 34.59 |  0.9176 | 29.39 | 0.8586 | 0.23s | 266.074G|
> | ReLoBlur* ($538\times359$)     | Restormer [1] | 33.10 | 0.8903  | 28.53 | 0.8376 | 0.26s | 450.210G |
> | ReLoBlur* ($538\times359$)   | Uformer [2] | 32.88 | 0.8747  | 28.38 | 0.8293 | 0.25s | 321.657G|
> | ReLoBlur* ($538\times359$)   | LMD-ViT (ours) | 33.52  |  0.8978 | 28.85  | 0.8451  | 0.10s |162.627G|
>
> [1] Zamir et al, Restormer, CVPR 2022
>
> [2] Wang et al, Uformer, CVPR 2022

---

> > ### Author Response · Authors · 2023-11-19
> > **Response to qustion 7 and minor issues**
> >
> > **Ablation study of no pruning and rest tricks.** In the original version of our paper, we discussed the performance without pruning strategy in Section 3.4 and Table 2. Line 5 in Table 2 are results of our proposed model without pruning strategies. Except for the pruning trick discussion, we discuss the effectiveness of blur mask annotation, W-LeFF, and feature channels in Section 3.5, Appendix E.1, and E.2, respectively. We add ablation studies of the other tricks in the new version of our paper, including the effectiveness of joint training in Appendix E.3, and local deblurring results under different resolutions in Appendix E.4, and the effectiveness of our backbone in Appendix E.5. All the modifications are highlighted in red.
> >
> > **About the minor issues.**
> >
> > 1) We delete the unfinished sentence and modified the first paragraph on page 2. We are very sorry for the previous mistake.
> >
> > 2) Thank you for your advice and we really appreciate it! We add the attention windows' borders in Figure 1 in the new version of our paper. Additionally, we deleted the visualization of AdaWPT 9 because the windows are too small to draw the borders. We add the visualization of AdaWPT 8 instead.
> >
> > 3) We change the first two points of the summarised contributions as: "1) the first sparse vision Transformer framework for local motion deblurring, LMD-ViT, which utilizes an adaptive window pruning strategy to focus computation on localized regions affected by blur and achieve efficient blur reduction without causing unnecessary distortion to sharp regions; 2) a sophisticated blurriness prediction mechanism, integrating a confidence predictor and decision layer to effectively distinguish between sharp and blurry regions". The modifications are highlighted in red in the new version of our paper.
> >
> > 4) We combine AdaWPT-F and AdaWPT-P as AdaWPT in Figure 2 of the new version of our main paper. Furthermore, we delete relative descriptions distinguishing them.
> >
> > 5) We replace Figure 6 with a more visible picture. The petals and wall tiles reconstructed with our pruning strategy are clearer than those without pruning.
> >
> > 6) Please refer to the reply to your fifth question. When inputting static sharp images, the masks are all zero, and the decision maps are nearly in all-black color. The results of these extreme experiments demonstrate the model’s ability to effectively distinguish between blurred and sharp images, showcasing our model's robustness. We add the results and discussion in the new version of Appendix E.7, highlighted in red.
> >
> > 7) In the domain of image restoration and deblurring, certain methods, exemplified by Li et al. [3] and Liang et al. [4], exhibit a spatially different treatment of regions. The paper "Mutual Affine Network for Spatially Variant Kernel Estimation in Blind Image Super-Resolution" is an example. The similarities between Liang et al. [4] and our proposed method are evident in two key aspects. Firstly, the motivation behind both of our methods involves performing distinct operations in different regions. Both approaches involve the spatially variant predictions for the adaptive reconstruction of degraded images. Liang et al. [4] estimate spatial variability kernels in the blind super-resolution task based on the degradation localizations. Similarly, our confidence predictor generates spatially variant confidence values for blurriness. Our proposed LMD-ViT treats local blurred regions and sharp regions differently, permitting only blur tokens to pass through Transformer layers. Secondly, both methods aim to constrain computation costs. Liang et al. [4] achieved this by leveraging channel inter-dependence without expanding the network's receptive field. In our approach, we perform deblurring and implement the pruning strategy in non-overlapping windows. This paper introduces multiple degenerate kernels in the problem of treating different regions differently, which is worthy of our learning and reference in future work. This discussion has been included in the second paragraph in Section 1. and Appendix A (related works). We cite Liang et al.'s paper [4] as a reference in the new version of our work.
> >
> > [3]Li et al., Real-World Deep Local Motion Deblurring, AAAI 2023
> >
> > [4] Liang et al., Mutual Affine Network for Spatially Variant Kernel Estimation in Blind Image Super-Resolution, Proceedings of the IEEE/CVF International Conference on Computer Vision 2021

---

### Official Review · Reviewer_SKAv · 2023-10-31

**Soundness:** 4 excellent
**Presentation:** 4 excellent
**Contribution:** 4 excellent
**Rating:** 6
**Confidence:** 4

**Summary:**

The presented work delves into an interesting research problem: local motion deblurring. It introduces a novel approach known as LMD-ViT, constructed using "Adaptive Window Pruning Transformer blocks (AdaWPT)." AdaWPT selectively prunes unnecessary windows, focusing powerful yet computationally intensive Transformer operations solely on essential windows. This strategy not only achieves effective deblurring but also preserves the sharp regions, preventing distortion. Moreover, their method significantly accelerates inference speed. Additionally, they have provided annotated blur masks for the ReLoBlur dataset. The utility of this approach is showcased on both local and global datasets, where it demonstrates substantial performance improvements on local motion deblurring (the ReLoBlur dataset) and competes favorably with baseline methods in the realm of global deblurring.

**Strengths:**

1.	This paper addressed problems of single image local motion deblurring, which is very essential in today’s photography industry. The presented method is the first to apply sparse ViT in single image deblurring and may inspire the community to enhance image quality locally.
2.	The proposed pruning strategy including the supervised confidence predictor, the differential decision layer and pruning losses are reasonable and practical. It combines window pruning strategy with Transformer layers, only allowing blurred regions to go through deblurring operations,  resulting in not only proficient deblurring but also the preservation of sharp image regions.
3.	The quantitative and perceptual deblurring performances are obvious compared to baseline methods.
4.	The presented method derives a balance between local motion deblurring performance and inference speed, as shown in the ablation study and experiments. The proposed method reduced FLOPs and the inference time largely without deblurring performances dropping on local deblurring data.
5.	The authors provided annotated blur masks for the ReLoBlur dataset, enhancing the resources available to the research community.

**Weaknesses:**

1.	The authors did not mention whether the presented method LMD-ViT requires blur mask annotation during inference. This is crucial because if the method does require blur masks before inference, it would be helpful to provide instructions on how to generate them beforehand and assess their practicality.
2.	The proposed method uses Gumble-Softmax as the decision layer in training and Softmax in inference. The equivalence of the two techniques in training and inference is not discussed.
3.	In the user study experiment, the absence of an explanation regarding the camera equipment used is notable. This is important because when images from the same camera share a common source, the blurriness often exhibits a consistent pattern. Therefore, including images from the same camera would allow us to assess the proposed method's robustness.
4.	Some references are missing, like “Window-based multi-head self-attention” in page 2, and “LeFF” in Section 2.4.1.

**Questions:**

please refer to the weaknesses.

---

> ### Author Response · Authors · 2023-11-18
>
> **Whether LMD-ViT requires blur masks in inference.** In the inference phase, the input consists solely of locally blurred images without accompanying blur masks. During training, blur mask annotations are utilized to guide the prediction of blurriness confidence. However, in the inference phase, our network is capable of autonomously predicting the locations of blurred regions without requiring additional blur mask annotations. We add this explanation in the second paragraph of Section 3.1, highlighted in red.
>
> **The equivalence of training and testing.** The difference between the training and inferring process lies in the decision layer. In inference, we use Softmax with a threshold to sparse the tokens. However, this technique is not suitable for training because setting a threshold to all images during training increases the instability of parallel training. Therefore, we use Gumbel-Softmax in training to overcome the non-differentiable problem of sampling from a distribution. These two methods have the same effect. Softmax produces a probability distribution over classes, while Gumbel-Softmax introduces stochasticity into the process of selecting discrete values in a differentiable way. Both of them sparse the tokens and distinguish between sharp and blurry tokens.
>
> **The details of acquiring real-world locally blurred images in the user study.** In the user study experiment, we acquired locally blurred images using a static Sony industrial camera and a static Fuji XT20 SLR camera. The camera sensors and shooting environments in this experiment differ from those in the ReLoBlur dataset and the GoPro dataset. Consequently, the blur patterns observed in the user study may exhibit variations compared to the training data. The visual results presented in Figure 4 demonstrate the robustness of our proposed model in effectively addressing real-world instances of local motion blur. We add the details in the third paragraph of Section 3.2, highlighted in red.
>
> **About the references.** Thank you for your advice. We have added the references of `"Window-based multi-head self-attention" on page 2, and "LeFF” in Section 2.4.1. The modifications are highlighted in red in the new version of our paper.

---

> ### Author Response · Authors · 2023-11-21
>
> Dear Reviewer SKAv:
>
> Thank you for taking the time to provide valuable feedback and constructive comments. Your advice has greatly helped us enhance the quality of our paper and proposed method. As we are nearing the end of the discussion period, we would like to inquire if our rebuttal has successfully addressed your concerns. If you have any additional questions, please feel free to respond to our responses. We are willing to address any remaining concerns you may have.
>
> Best Wishes,
>
> Authors of paper "Adaptive Window Pruning for Efficient Local Motion Deblurring"

---

### Official Review · Reviewer_ePoq · 2023-10-31

**Soundness:** 2 fair
**Presentation:** 2 fair
**Contribution:** 2 fair
**Rating:** 3
**Confidence:** 5

**Summary:**

The paper attempts to tackle local motion deblurring. Existing deblurring literature mostly focuses on general deblurring (without specifying global or local regions). This could be a disadvantage in computation and generalization in scenarios where only a small part of a high-resolution image is blurred. Therefore, this work proposes a transformer-based network called LMD-ViT. The authors make use of adaptive window pruning and blurriness confidence predictor in the transformer blocks to ensure the network focuses on local regions with blurs. Quantitative and qualitative results are presented on the ReLoBlur and GoPro datasets. The effectiveness of different design choices is analyzed.

**Strengths:**

An adaptive window pruning strategy is adopted to focus the network computation on localized regions affected by blur and speed up the Transformer layers.

A carefully annotated local blur mask is proposed for the ReLoBlur dataset to improve the performance of local deblurring methods.

**Weaknesses:**

The organization of the paper can be improved.

1) The methodology (Sec. 2) consists of too many (unnecessary) acronyms. Moreover, there are some inconsistencies when citing previous works (for example, LBAG (Li et al., 2023), LBFMG (Li et al., 2023), etc.). It would be better for the submission would strongly benefit from polishing the writing.

The settings of the experiments need more explanation.

2) It is not clear why the GoPro dataset is used for training along with the ReLoBlur training set. In previous works, such as LBAG, only the ReLoBlur dataset is used (see Table 4).

The novelty of the submission needs to be clarified.

3) It would be better to discuss the differences between LBAG and Uformer. Compared to LBAG, it simply substitutes the CNN architecture with a Transformer. All the other modules, including sparse ViT, W-MSA (Window-based multi-head self-attention), and LeFF (locally- enhanced feed-forward layer), have been introduced in previous deblurring works.

The fairness of the experiments.

4) The transformer baselines, such as Restormer and Uformer-B, are not trained with the local blur masks, which are deployed during their training by the proposed methods. This makes the comparison in Table 1 unfair.

Unclear parts.

5) Please explain in detail how the authors manually annotate the blur masks for the ReLoBlur dataset.

6) The baseline results reported in Table 1 are higher than those in their original papers, e.g., LBAG. It would be better to give the reasons and more details when introducing Table 1. The baseline results reported in Table 1 are higher than those in their original papers, e.g., LBAG for 34.85 dB.

Typo:
Table table 4 in Appendix C.
Decision map in Figure 7.

**Questions:**

See Weaknesses.

---

> ### Author Response · Authors · 2023-11-18
>
> **About the organization and writing.** Thank you for your advice. To improve the writing of our paper, we have made several modifications:
>
> 1) We have removed the second paragraph from Section 1 and incorporated discussions about our novelty in the same section.
>
> 2) We have consolidated the explanations for AdaWPT-F and AdaWPT-P under a unified term, AdaWPT, in Section 2. Accordingly, we've updated Figure 2 to reflect this change without differentiating between AdaWPT-F and AdaWPT-P.
>
> 3) We add more discussions in the new version of our Appendix. Specifically, the blur mask annotation details in Appendix C, the effectiveness of the joint training technique in Appendix E.3, deblurring results under different resolutions in Appendix E.4, the effectiveness of our backbone in Appendix E.5, mask prediction accuracy in Appendix E.6, results of special cases in Appendix E.7, as suggested by the reviewers.
>
> 4) To maintain the consistency of citing LBAG [1], we replace "LBFMG" [1] with "LBAG" [1] in the new version of our paper. (Notably, LBFMG [1] is a blur mask generation method in paper LBAG. In the original version of our paper, we wrote LBFMG [1] when discussing blur mask annotations.)
>
> All the modifications are highlighted in red in the new version of our paper.
>
> **Why we use the GoPro [2] and ReLoBlur dataset [1] for joint training.**
> We train with the GoPro dataset [2] and ReLoBlur dataset [1] together mainly because of two reasons. Firstly, joint training could prevent our model from over-fitting and improve the model's robustness. Secondly, we expect our proposed LMD-ViT to deblur both globally and locally. Training with the GoPro [2] and ReLoBlur [1] datasets together improves both the local and global deblurring performances. We compare models trained solely on the ReLoBlur dataset [1] and jointly trained with the two datasets in the Table below. The results show improvements (+0.13dB in local deblurring and +0.42dB in global deblurring) when we add the GoPro dataset [2] to train. We've added this discussion to the new version of Appendix E.3.
>
> | Training data | Training data          | Testing data | Testing data |
> | ------- | ------- |------- | ------- |
> | ReLoBlur      | ReLoBlur & GoPro | ReLoBlur      | GoPro |
> |          ✓         |                              |   35.29 dB / 0.9280 |   31.74 dB / 0.9271  |
> |                      |              ✓             | 35.42 dB / 0.9285   |   32.16 dB / 0.9318  |
>
> [1] Li et al, Real-World Deep Local Motion Deblurring, AAAI 2023
>
> [2] Nah et al, Deep multi-scale convolutional neural network for dynamic scene deblurring, CVPR 2017

---

> > ### Author Response · Authors · 2023-11-18
> > **Response for the third question**
> >
> > **The difference between LBAG [1] and our method.**
> >
> > 1) LBAG [1] uses a gate structure at the end of its network to mitigate the deblurring impact on non-blurred regions. However, it still involves unnecessary computations as the entire image has already been processed by the previous parts of the network network. Our proposed LMD-ViT predicts blur regions and prunes tokens at the very start. Thus, it saves the computation for Transformers as much as possible.
> >
> > 2) LBAG [1] is a CNN-based method. In LBAG [1], the interactions between the image and convolution kernels are content-independent and ill-suited for modeling long-range dependencies. We discussed these two differences in the $2^{nd}$ paragraph of Section 1.
> >
> > 3) We manually annotate blur mask ground truths rather than using that from LBAG [1]. This is because the masks produced by LBAG [1] have holes and noise, which decrease the precision of window pruning. On the contrary, our annotations are smooth and more in line with human vision. They improve the blur detection precision of our methods and other methods, as illustrated in Section 3.5 of our main paper.
> >
> > **The difference between Uformer [3] and our method.**
> >
> > 1) Unlike Uformer [3], which conducts attention operations on every token and consequently distorts sharp tokens in locally blurry images (as illustrated in Figure 1 of our main paper), while also wasting computational resources during inference (as demonstrated in Table 1 of our main paper), our method is more efficient: it filters out unnecessary tokens, allowing only the blurry ones to pass through the Transformer layers. This approach significantly reduces computation and is ideally suited for local motion deblurring tasks. We have included this discussion in the second paragraph of Section 1, highlighted in red, in the updated version of our paper.
> >
> > 2) Different from the main block of Uformer [3] which focuses on restoring images globally, the main block of our method, AdaWPT, prunes tokens based on a confidence predictor and a decision layer trained by a reconstruction loss with Gumbel-Softmax re-parameterization and a pruning loss guided by annotated blur masks.
> >
> > **The novelty of our work.** Our novelty lies in three aspects.
> > *Firstly,* the key idea of our method lies in pruning redundant tokens to protect sharp regions as well as enhance inference speed for local motion deblurring Transformers.
> > *Secondly,* our work is the first to apply sparse ViT in image deblurring. We introduce a novel AdaWPT block, which only allows windows with blurry regions to go through W-MSA, leveraging the blurriness confidence predicted by an end-to-end trained confidence predictor. The training process takes advantage of a reconstruction loss with Gumbel-Softmax re-parameterization and a pruning loss guided by annotated blur masks. By doing so, we significantly reduce inference time by half and enable the network to concentrate on locally blurred regions. Though sparse ViT was previously adopted in the image classification task, they used a top-K strategy to decide how many tokens should be abandoned in each layer. However, this restricts the model expressions in different locally blurred images. Our proposed method selects tokens with predicted blur mask and performs natural and reasonable token pruning.
> > *Thirdly,* to better adapt LeFF to local motion deblurring, we introduce a modified version called W-LeFF. W-LeFF only performs convolution within blurry windows. Experiments in Appendix D.1 demonstrate that our proposed W-LeFF enables parallel training and faster inference without significantly compromising performance.
> > *In conclusion,* we have developed an innovative adaptive window pruning Transformer for local motion deblurring that is both effective and efficient. It is the first approach to apply sparse ViT to image deblurring, effectively bridging the gap between efficient local motion deblurring and ViT. Compared to the CNN-based deblurring method, we achieved improved visual and quantitative performances. Compared to Transformer-based methods, we speed up inference. Though we borrow W-MSA from Uformer, we do not regard it as our contribution. We summarize this discussion in the $3^{rd}$ paragraph in Section 1 in the new version of our paper, highlighted in red.
> >
> > [1] Li et al, Real-World Deep Local Motion Deblurring, AAAI 2023
> >
> > [3] Wang et al, Uformer: A general u-shaped transformer for image restoration, CVPR 2022

---

> > > ### Author Response · Authors · 2023-11-18
> > > **Response to the fourth to seventh question**
> > >
> > > **The fairness of the experiments.**  To utilize the blur masks to train, we borrow the gated block from LBAG [1], and the weighted loss from our method (introduced in Section ). We apply the two local blur-aware strategies to Uformer [3] and Restormer [4]. The following table shows that when trained with local blur masks, the PSNR and SSIM scores of Uformer [3] and Restormer [4] improve. However, they need more time to infer and the overall performances are still inferior to ours without the pruning strategy and AdaWPT blocks. We've added this discussion to the new version of Appendix E.5.
> > >
> > > | Methods       | PSNR | SSIM | PSNR\_w | SSIM\_w| Inference time|
> > > |-----------------|----------|----------|-----------------|----------|----------|
> > > |Restormer w/ mask| 34.88 | 0.9259 | 29.44 | 0.8806 | 3.75s |
> > > | Uformer-B w/ mask| 35.23 | 0.9277 | 30.22 | 0.8935 | 1.33s |
> > > | LMD-ViT (ours)           |35.42 | 0.9285 | 30.25 | 0.8938 |0.56s |
> > >
> > > In the original version of our paper, we did not train baselines with local blur masks for two reasons: 1)  we follow the comparison settings of the LBAG paper, which also trains the baseline networks without blur mask annotations; 2) we want to avoid altering the carefully designed network structures of baseline methods. Noticeably, we aim to compare the baselines in a fair and equitable manner without compromising any structure. Being able to use the blur mask is precisely the motivation and advantage of our model. We regard the whole strategy as our novelty rather than the backbone only.
> > >
> > > **The details of blur mask annotations.** As illustrated in Section 2.4.2 of our main paper, we manually mark binary masks for the locally blurred images of the ReLoBlur dataset. In our annotated blur masks, pixels with a value of 1 indicate blurriness, while pixels with a value of 0 represent sharpness. This can be seen in Figure 1 and Figure 5 of our main paper. To confirm the blurry regions, we subtract the sharp and blurred images to observe the differences in local regions. This difference map helps us to determine the approximate position of the blurriness and the border. After confirming the blurry regions, we select moving objects on the \textit{EISeg} software, which can automatically segment different components within each object. Then, we manually refine the blurry regions. To prevent leakage, we extend the blurry region by 5 pixels, ensuring that all pixels within the annotated mask are considered blurry. Regarding pixels between blurry and sharp areas, we adhere to a principle: a diffuse spot encompassing more than 5 pixels is categorized as a blurry region, while conversely, it is classified as a sharp region. We add this discussion in the new version of Appendix C.
> > >
> > > **The reason why our reported scores are higher than that of the LBAG [1] paper.** There are mainly three factors causing the score difference. 1) We use our newly annotated blur masks for supervised training rather than the blur masks provided by LBAG [1] in their original paper, which improves the LBAG's score [1]. 2) We train LBAG [1] with the ReLoBlur dataset [1] and the GoPro dataset [2] together. 3) We train LABG [1] for 160k steps in our paper. The results from different training steps may fluctuate a little, using the official public code of LBAG [1]. Due to the above three reasons, the scores are higher than those reported in the LBAG [1] paper.
> > >
> > > **About the typos.** Thank you for your advice. We have replaced "Gopro" with ``GoPro" in Table 4 in Appendix C. We modified the sentences in Figure 7: " All the values in the decision map are equal to 1". All the modifications are highlighted in red.
> > >
> > > [1] Li et al, Real-World Deep Local Motion Deblurring, AAAI 2023
> > >
> > > [2] Nah et al, Deep multi-scale convolutional neural network for dynamic scene deblurring, CVPR 2017
> > >
> > > [3] Wang et al, Uformer: A general u-shaped transformer for image restoration, CVPR 2022
> > >
> > > [4] Zamir, Restormer: Efficient transformer for high-resolution image restoration, CVPR 2022

---

> ### Author Response · Authors · 2023-11-21
>
> Dear Reviewer ePoq:
>
> Thank you for taking the time to provide valuable feedback and constructive comments. As we are nearing the end of the discussion period, we would like to inquire if our rebuttal has successfully addressed your concerns.
>
> If you have any additional questions, please feel free to respond to our responses. We are more than willing to address any remaining concerns you may have. Once again, thank you for all the helpful comments provided.
>
> Best Wishes,
>
> Authors of paper "Adaptive Window Pruning for Efficient Local Motion Deblurring"

---

> > ### Comment · Reviewer_ePoq · 2023-11-23
> >
> > Dear authors,
> >
> > Thank you for your response.
> >
> > I would like to clarify whether Restormer (Zamir et al., 2022) and Uformer-B (Wang et al., 2022) in Table 1 and Table 9 were trained on the same dataset as the proposed method.
> >
> > It is difficult to determine whether the performance improvement of the proposed method is due to the extra training data (ReLoBlur & GoPro) or the extra label (mask) based on the current version.
> >
> > Table 4 is a relatively fair comparison. All methods were trained with the same training set and tested on the GoPro dataset, where the proposed method achieved the second-best performance. This also raises questions about the benefits of the proposed framework that using masks. In other words, while the proposed framework can benefit local blur, it restricts the performance on global blur.
> >
> > In summary, I still have concerns about the fairness of the experiment and the performance of the proposed method.
> >
> > Best regards

---

> ### Author Response · Authors · 2023-11-23
>
> Dear Reviewer ePoq,
>
> Thank you for your inquiry.
>
> Firstly, all the baselines, including Restormer [3] and Uformer-B [4] in Table 1 and Table 9 were trained on the same dataset as our proposed method.
>
> For your second concern **whether the performance improvement of the proposed method is due to the extra training data or the extra label**, we think the performances are improved for both of the two reasons. We've done experiments before with controlled variables. Firstly, we train our method with the GoPro dataset [2] and the ReLoBlur dataset [1] together with our newly annotated masks, like the results presented in response to your second question. In our previous response regarding "Why we use the GoPro [2] and ReLoBlur dataset [1] for joint training", experiments No.1 and No.2 share the same training settings, differing only in the inclusion of the GoPro dataset [2]. Therefore, the results in our previous response substantiate that joint training can enhance deblurring performance. Secondly, the verify the influences of blur masks, we have included the results of training with masks annotated by the LBFMG [1] method in the updated table below. The results show that our newly annotated blur masks also contribute to the enhancement. Moreover, the updated table could further strengthen the assertion that training with both the ReLoBlur dataset [1] and the GoPro dataset [2] is a superior choice compared to training solely with the ReLoBlur dataset [1]. We have incorporated these updated findings in Table 7 of the new version.
>
> | Training data | Training data| Blur mask | Blur mask | Testing data | Testing data |
> | ------- | ------- | ------- | ------- | ------- | ------- |
> | ReLoBlur    | ReLoBlur +GoPro   | LBFMG's   | Ours  | ReLoBlur              | GoPro|
> |✓		     |                                |✓                 |           |35.24 dB / 0.9283 |31.60 dB / 0.9252 |
> |                    |✓	                      |✓                 |           |35.29 dB / 0.9280 |31.74 dB / 0.9271
> | ✓                |                                 |                   | ✓        |35.31 dB / 0.9270 |31.14 dB / 0.9217 |
> |                    |   ✓                           |                   | ✓        |35.42 dB / 0.9285 |32.16 dB / 0.9318 |
>
> For your third concern of **fairness** and **global deblurring performances**, we ensure that all experiments are conducted fairly and traceably. We provided the results of training with baselines and extra data (blur mask) in the previous response and illustrated that with the same information, our model achieves the best local deblurring performance.
>
> Our module design is focused on local deblurring, that is, how to adaptively remove some areas that do not participate in the calculation while ensuring that there is a local performance improvement. From Table 1 in our main paper, we can see that our proposed method LMD-ViT has achieved great advantages while reducing a lot of computation. Table 4 shows the comparable performance of global deblurring and makes perfect sense, but global deblurring is not the focus of our design module concerns.
>
> If our response addresses your concerns adequately, could you please update the rankings of our paper? Thank you for your time and interest in our work!
>
> Best wishes,
>
> Authors of paper "Adaptive Window Pruning for Efficient Local Motion Deblurring"
>
> [1] Li et al, Real-World Deep Local Motion Deblurring, AAAI 2023
>
> [2] Nah et al, Deep multi-scale convolutional neural network for dynamic scene deblurring, CVPR 2017
>
> [3] Wang et al, Uformer: A general u-shaped transformer for image restoration, CVPR 2022
>
> [4] Zamir, Restormer: Efficient transformer for high-resolution image restoration, CVPR 2022

---

### Author Response · Authors · 2023-11-23
**Kind reminders: The summary of our modifications and the link to our codes**

We appreciate all the reviewers for their valuable time and constructive comments. It enables us to substantially improve the quality of our paper and proposed dataset.

In our revised paper, all the issues that the reviewers raised have been addressed. The changes we made can be summarized as follows:

1) To evaluate the effectiveness of our proposed method LMD-ViT, we add more ablation studies in the Appendix, including the effectiveness of the joint training techniques (Appendix E.3), local deblurring results under different resolutions (Appendix E.4), the effectiveness of our backbone (Appendix E.5), mask prediction accuracy (Appendix E.6), and special cases (Appendix E.7).

2) We have added more blur mask annotation details in Appendix C.

3) We have polished our writing and improved the organization of our paper. For instance, we added grids in Figure 1 to denote window borders; we united AdaWPT-F and AdaWPT-P as the AdaWPT layer in Figure 2, and we corrected some typos. All the modifications are highlighted in red.

4) We added more references related to our paper (i.e. "Mutual affine network for spatially variant kernel estimation in blind image super-resolution") and corrected the reference mistakes (i.e. LeFF, LBFMG).

5) We have provided an anonymous link to our code:
https://drive.google.com/drive/folders/1t7Nkhh8cNQeCvxgHTG-DAw1ze7QWQ0ju?usp=sharing.

Since it is close to the end of the discussion period, we would like to kindly remind the reviewers to confirm whether our changes solved your concerns. If you have any further questions, do not hesitate to reply to our responses. Without your help, we could not build this useful, reasonable, and robust dataset. Thank you again for all the efforts that you have made.

Best wishes,
Authors of paper "Adaptive Window Pruning for Efficient Local Motion Deblurring"

---

### Meta-Review · Area_Chair_UZou · 2023-12-10

**Metareview:**

This paper tackles dynamic scene deblurring problem by applying an adaptive pruning strategy to a transformer-based neural network.

The biggest concern from the review was the fairness of the comparisons as different training datasets could make the experimental results inconclusive. In the rebuttal, the authors provided experimental results by different training datasets and it shows the proposed method improves deblurring accuracy from fixed training data.

Acceptance is recommended. Please include as many details from the rebuttal discussion in the final manuscript.

**Justification For Why Not Higher Score:**

Although the concerns from the reviewers are addressed, there were many agendas and the fix is not fully reflected in the manuscript, yet.
The authors should do additional works to improve the quality of the paper.

**Justification For Why Not Lower Score:**

From the intense discussion between the authors and the reviewers, the concerns are well addressed.

---

### Decision · Program_Chairs · 2024-01-16

Accept (poster)